# CNN Kernels Can Be the Best Shapelets

**Eric Qu**[1], **Yansen Wang**[2], **Xufang Luo**[2], **Wenqiang He**[3], **Kan Ren**[4], **Dongsheng Li**[2]
[1]University of California, Berkeley, [2]Microsoft Research Asia,
[3]University of Science and Technology of China, [4]ShanghaiTech University
`ericqu@berkeley.edu,`
`{yansenwang,xufluo,dongsheng.li}@microsoft.com,`
`wenqianghe@mail.ustc.edu.cn,renkan@shanghaitech.edu.cn`

## Abstract

Shapelets and CNN are two typical approaches to model time series. Shapelets aim at finding a set of sub-sequences that extract feature-based interpretable shapes, but may suffer from accuracy and efficiency issues. CNN performs well by encoding sequences with a series of hidden representations, but lacks interpretability. In this paper, we demonstrate that shapelets are essentially equivalent to a specific type of CNN kernel with a squared norm and pooling. Based on this finding, we propose ShapeConv, an interpretable CNN layer with its kernel serving as shapelets to conduct time-series modeling tasks in both supervised and unsupervised settings. By incorporating shaping regularization, we enforce the similarity for maximum interpretability. We also find human knowledge can be easily injected to ShapeConv by adjusting its initialization and model performance is boosted with it. Experiments show that ShapeConv can achieve state-of-the-art performance on time-series benchmarks without sacrificing interpretability and controllability.

## 1 Introduction

In the realm of machine learning, interpretable time-series modeling stands as a pivotal endeavor, striving to encode sequences and forecast in a manner that resonates with human comprehension. Among an array of early methods to distill interpretable features from sequences, *shapelets* (Ye & Keogh, 2009) have garnered significant attention, finding applications in diverse downstream tasks. These shapelet are discriminative sub-sequences culled from the primary time series and the minimal distance between a shapelet and all conceivable sub-sequences of the raw input is ascertained, yielding features that signify a shapelet's imprint on a sequence. The allure of shapelets lies in their capacity to discern local discriminative patterns inherent in the data. However, conventional shapelets grapple with inefficiencies, attributed to their exhaustive search demands and elevated time complexity.

As the new era of deep learning comes, more and more works seek to fit the sequence with a high dimensional non-convex function using deep neural networks such as RNN (Guo et al., 2019), CNN (Franceschi et al., 2019), Transformer (Wu et al., 2021; Qu et al., 2022; Cheng et al., 2023), etc. to model the time series. These *deep-learning-based methods* have attracted much more attention than shapelets, thanks to their great performance when the number of data is sufficient, but they are more likely to overfit when the signal-to-noise ratio is relatively low and the data are scarce. Also, the representations (often called hidden representations) are almost impossible to interpret and control due to the black-box nature of neural networks. While subsequent research endeavors (Ma et al., 2020b; Li et al., 2022; He et al., 2023) are proposed aiming at fusing the interpretability of shapelets and the promising performance of deep methods, they often fail with striking a harmonious equilibrium between performance and interpretability.

In this paper, we aim to seamlessly inject the interpretability of shapelets into the convolutional layer while retaining the advantages and characteristics of both. Despite the apparent disparity between shapelets and deep models in time-series modeling, for the forward process, we first theoretically prove that *extracting features with shapelets can be equivalently conducted by passing the input time*

---

This work was done during Eric Qu and Wenqiang He's internship at MSRA. Correspondence to: Xufang Luo (xufluo@microsoft.com; Dongsheng Li (dongsheng.li@microsoft.com).

*series to a specific convolutional layer (1-layer CNN) with a squared norm and pooling.* This finding provides us the basis for combining shapelets with deep models.

To implement the equivalence, we devise **ShapeConv**, a CNN layer wherein its kernel functions as shapelets, adeptly and interpretably addressing the time-series modeling challenge. We introduce several ingenious designs to make ShapeConv effective in practice. During the optimization process, due to the difference of candidate space, the subsequences derived via gradient-based techniques might diverge significantly from the sub-sequence prototypes, rendering them less suitable as interpretable shapelets. Hence, ShapeConv incorporates an additional shaping regularization to enforce similarity. Besides, another regularization term is utilized to relieve the issue that the model tends to fall into the local optimal point where kernels are similar but not catching diverse and discriminative features. As for the initialization, we also design separate judicious strategies to make model weights close to different discriminative sub-sequences in data in different tasks, capturing class-specific and cluster-specific information for supervised classification and unsupervised clustering, repsectively.

In contrast to traditional shapelet techniques, ShapeConv, being a deep model, facilitates end-to-end optimization. This paves the way for parallel computing for acceleration, effortless stacking with deep modules for improved performance. When compared to learning-based approaches, our kernels provide controllability, facilitated by our strategic initialization that incorporates human expertise to yield more human-comprehensible results. Empirical evaluations underscore ShapeConv's prowess in both supervised classification and unsupervised clustering tasks and various datasets. It surpasses other learning-based shapelet techniques and contemporary deep models tailored for time-series classification and clustering, all the while preserving interpretability. Furthermore, the infusion of human knowledge amplifies the model's performance, and there's a marked reduction in time complexity when compared to earlier shapelet methodologies.

We summarize our contribution as 4 folds: *(1)* We have formally and theoretically proven the equivalence of a specific CNN layer, when combined with square norm and pooling, to the shapelet. *(2)* Based on the discovered equivalence, we introduce ShapeConv, an interpretable CNN layer with its kernel serving as shapelets. Several regularizations and initializations are accompanied to enforce similarity and diversity, making ShapeConv effective in practice. *(3)* By treating the CNN kernel as a shapelet, we claim another advantage of facilitating the incorporation of human prior knowledge by initialization. *(4)* Extensive experiments on real-world datasets validates ShapeConv's superior performance on interpretable time-series modeling tasks.

## 2    BACKGROUND

**Shapelets**    Shapelet (Ye & Keogh, 2009), originally defined as a maximally discriminative sub-sequence in time-series data (shown in Figure 1), is designed to capture inter-class features in terms of small sub-sequences rather than the full sequence. Shapelet-based methods will usually find a subset of shapelets $S^* \subset \hat{S}$ from data to maximize information gain by splitting data with the shapelets as nodes of a decision tree. The candidate shapelet set $\hat{S}$ contains all possible sub-sequences of the original data.

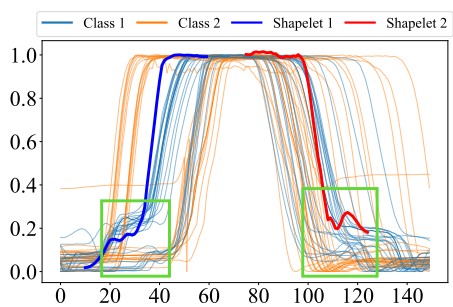

Afterwards, a shapelet transform (Lines et al., 2012) is introduced to decouple the shapelet discovering step and downstream classifier. This shapelet transform step will extract features for an input signal $\mathbf{X}$ based on the distances between shapelets and the input. Specifically, for each selected shapelet $\mathbf{s} \in S^*$, we calculate the minimal distance between $\mathbf{s}$ and all sub-sequences of $\mathbf{X}$, i.e.,

Figure 1: An example of time-series classification task and learned shapelets by ShapeConv. Both shapelets are for class 1, and they capture two distinguish shapes of the class, respectively (indicated by green frames). Lines of class 1 exhibit more pronounced fluctuations at these two locations.

$$d_{\mathbf{s},\mathbf{X}} = \min_{\mathbf{x} \in \hat{\mathbf{X}}} dist(\mathbf{s}, \mathbf{x}), \qquad (1)$$

where $\hat{\mathbf{X}}$ is the set containing all sub-sequences with the same length as $\mathbf{s}$, and $dist(\mathbf{s}, \mathbf{x})$ is the distance function (usually the squared Euclidean distance) measuring the similarity between a shapelet and a raw sub-sequence. Using all shapelets in $S^*$, the feature vector with length $|S^*|$ can be obtained for each $\mathbf{X}$, and these features are used for building different kinds of classifiers besides tree-based models. We provide a more vivid example on shapelets and shapelet transform in Appendix B.2.

Beyond traditional costly shapelet discovering methods, following works extend the candidate set to $\mathbb{R}^{l_s}$ where $l_s$ is the length of shapelet, and try to solve the problem with optimization-based methods.

**Convolutional Neural Networks**   CNN is one of the most commonly used network structures and its several variants have been used in time-series modeling for years (Ismail Fawaz et al., 2019). A CNN layer is essentially some sliding filters known as convolutional kernels followed by the activation function and pooling layer. Here, we consider a simple case to illustrate the CNN layer, where the input is 2-dimensional signal $\mathbf{X} \in \mathbb{R}^{n_{in} \times l_x}$ with $n_{in}$ input channels and $l_x$ time steps. Applying a 1-D convolution over it can be formulated as:

$$\mathbf{Y} = \sum_{k=1}^{n_{in}} \mathbf{W}_k * \mathbf{X}_k, \tag{2}$$

Where $\mathbf{W} \in \mathbb{R}^{n_{in} \times n_{out} \times l_s}$ denotes the weights of convolutional kernels. The $*$ symbol denotes the cross-correlation operator on the $k$-th input channel between each raw vector of $\mathbf{W}_k$ and the input $\mathbf{X}_k$. Note that for most CNN kernels, weights $\mathbf{W}_k$ is learnable so this cross-correlation is equivalent to the convolution in terms of optimization. After that, a non-linear activation function and a pooling layer are often applied to extract the aggregated value among its neighbors $N(j)$ for each channel $i$ at location $j$, which can be written as:

$$\mathbf{Y}_{ij}^{pool} = \underset{j' \in N(j)}{\text{pool}} \; \sigma(\mathbf{Y}_{ij'} + \mathbf{b}_i). \tag{3}$$

Here, The bias term $\mathbf{b}$ is sometimes set to zero for simplicity. The symbol $\sigma$ denotes the activation function such as ReLU, Tanh, and $\text{pool}$ denotes a pooling function such as $\max$ or $\text{mean}$.

## 3   HOW CAN CNN KERNELS BE THE BEST SHAPELETS?

In this section, we give a comprehensive answer to the question in the title. First, we provide a formal proof to show the equivalence between CNN and shapelets in Sec. 3.1. Then, we introduce ShapeConv, a novel convolutional layer well utilizing the equivalence in Sec. 3.2. Finally, we show how ShapeConv can be used for supervised learning and unsupervised learning in Sec. 3.3 and Sec. 3.4, respectively.

### 3.1   EQUIVALENCE BETWEEN CNN KERNELS AND SHAPELETS

The core idea is that when the calculation of squared Euclidean distance in the shapelet transform step is expanded, one of the terms is exactly the same as the forward passing in a CNN layer. Therefore, using these shapelets to extract features with squared Euclidean distance for $\mathbf{X}$ can be equivalently done by convolving $X$ with $n_{out}$ kernels from a 1-D CNN layer added by squared $L_2$ norm, followed by a maximum pooling. The difference between these two can be easily handled and omitted in practice. We summarize the finding in the following theorem:

**Theorem 3.1** *Assume the input $\mathbf{X} \in \mathbb{R}^{l_x}$ is a 1-dimensional single-variate signal of length $l_x$, and $n_{out}$ shapelets $S^* = \{\mathbf{s_1}, \mathbf{s_2}, ... \mathbf{s_{n_{out}}}\}$ with length $l_s$ are discovered. The feature extracted from $\mathbf{X}$ with $\mathbf{s_i}$ and squared Euclidean distance is $d_{\mathbf{s_i}, \mathbf{X}}$. Then we have*

$$d_{\mathbf{s_i}, \mathbf{X}} = -2 \max_{j \in \{1, 2, ..., l_x - l_s + 1\}} \left[ \mathbf{Y}_{ij} - \mathcal{N}(\mathbf{s_i}, \mathbf{X}_{j:j+l_s-1}) \right], \tag{4}$$

*where $\mathbf{Y} = \mathbf{s_i} * \mathbf{X}$ is the cross-correlation defined in Eq. 2 and $\mathcal{N}(\mathbf{s_i}, \mathbf{X}') = (\|\mathbf{s_i}\|_2^2 + \|\mathbf{X}'\|_2^2)/2$ is squared $L_2$ norm term.*

Detailed proof can be found in the Appendix C. Eq. 4 bridges shapelets and the learnable CNN kernel. The left-hand side is the feature extracted by a specific shapelet, and the right-hand side contains the

maximum pooling over the convolution between the kernel and the input time series and a squared norm term. The difference between these two is a constant factor -2 which can be absorbed in the learnable parameters.

There are two more gaps between the practical use of CNN in Eq. 3 and Theorem 3.1. One is the non-linear activation function. When the activation function $\sigma$ is monotonically increasing, the order to apply maximum pooling and activation function can be swapped, i.e., $\max(\sigma(\cdot)) = \sigma(\max(\cdot))$. This fits for most cases in practice with ReLU, Tanh, Sigmoid and their variants as activation functions. Another is the bias term. Since $\mathbf{b}$ is often designed to be independent of the position, we can have $\max_j(\mathbf{Y}_{ij} + \mathbf{b}_i) = \max_j(\mathbf{Y}_{ij}) + \mathbf{b}_i$. Now we can rewrite Eq. 3 as:

$$\mathbf{Y}_{ij}^{max} = \max_{j' \in N(j)} \sigma(\mathbf{Y}_{ij'} + \mathbf{b}_i) = \sigma(\max_{j' \in N(j)} (\mathbf{Y}_{ij'}) + \mathbf{b}_i). \tag{5}$$

This suggests that we can optionally add the bias term and the non-linear activation after the features are extracted with the shapelets to obtain a complete equivalence.

## 3.2 SHAPECONV: AN INTERPRETABLE CNN LAYER WITH ITS KERNELS SERVING AS SHAPELETS

Motivated by the above established equivalence, we introduce ***ShapeConv***, a novel interpretable CNN layer for time-series data, with its kernels serving as shapelets. Specifically, in addition to the cross-correlation operator and max pooling in the original CNN layer, we add a squared norm term $\mathcal{N}(\mathbf{s_i}, \mathbf{X}')$ in Eq. 4 before the pooling function to the CNN layer. Consequently, ShapeConv's forward pass mirrors the shapelet transform, calculating the minimum distance between a shapelet and all possible raw input sub-sequences, and convolutional kernels in ShapeConv play the same role as shapelets. Although initially univariate, ShapeConv can directly adapt to multivariate data tasks by adjusting the number of input channels while maintaining its other designs, enabling the model to learn multiple kernels/shapelets for each variate.

**Shaping Kernels** To serve as a good shapelet, ShapeConv's kernel should be the maximally discriminative thus can provide solid criteria for downstream classification (discriminability), and as human-comprehensible as possible by the minimize the distance from the sub-sequence of the data (interpretability).

The discriminability is achieved by optimizing kernel weights via task-specific loss which will be discussed in Sec. 3.3 and Sec. 3.4, respectively. As for the interpretability, while extending the candidate set from $S^*$ to $\mathbb{R}^{l_s}$ allows the learning-based methods to achieve best classification results, shapelets which look way too different from the input data conversely downgrade the overall interpretability. Therefore, to shape kernels like original data, we first strategically design the initialization method. Depending on whether class labels are available, we suggest different methods for initial kernel-data proximity, elaborated in the following subsections. Besides, we introduce a shape regularizer to keep the kernel similar to data during training. Specifically, we calculate distances between the kernel weight, i.e., shapelet $\mathbf{s_i}$, and sub-sequences in the input, and the shape regularizer is defined as the minimal distances, i.e.,

$$\mathcal{R}_{shape} = \frac{1}{n_{out}} \sum_{i=1}^{n_{out}} \min_{\mathbf{x} \in \hat{\mathbf{X}}_i} dist(\mathbf{s_i}, \mathbf{x}), \tag{6}$$

where $\hat{\mathbf{X}}_i$ is the set containing all sub-sequences with same length as $\mathbf{s_i}$. We opt for squared Euclidean distance here to allow for the reuse of the distance calculated in the shapelet transform step. This approach ensures kernel weights are initialized with original data shapes and remain close during training, yielding interpretable kernels with discriminative shapes.

**Increasing Diversity** Since shapelets in our model are learnable weights with large flexibility, they tend to fall into local optimality where all kernels are similar but not catching diverse and discriminative features. We provide experimental results on this phenomenon in Appendix D.3. Therefore, we further introduce a diversity regularizer following Zhang et al. (2018) to relieve this issue. We first use the pairwise $\ell_2$ distance between different kernels (or shapelets) to construct a distance matrix $\mathbf{D}_s$, where $\mathbf{D}_s(i,j) = \exp(-\ell_2(\mathbf{s_i}, \mathbf{s_j}))$, and the diversity regularizer is defined as $F$-norm of $\mathbf{D}_s$, i.e.,

$$\mathcal{R}_{div} = \|\mathbf{D}_s\|_F. \tag{7}$$

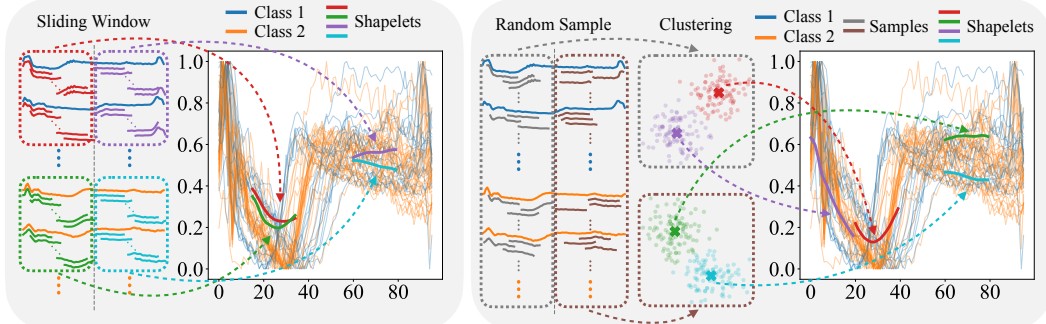

Figure 2: Illustration of initialization methods for ShapeConv kernels. We let $n_{out} = 4$, $n_{cls} = 2$ for both figures. Left part shows initialization in supervised learning, $k = 2$ here. Right part shows initialization in unsupervised learning, $n_{cut} = 2$ and $k = 2$ here.

To summarize, the above basic designs make ShapeConv a differentiable CNN layer with its kernels serving as shapelets. ShapeConv can keep the advantages of both CNN and shapelets. On the one hand, ShapeConv is a fully differentiable CNN layer, so it can be optimized effectively with lower time complexity than traditional methods. On the other hand, kernels of ShapeConv have good interpretability since they are similar to discriminative sub-sequences in original data.

## 3.3 SUPERVISED LEARNING

In this section, we show the method to apply ShapeConv in a supervised learning task. Here, we focus on the time-series classification task, which is a typical application for shapelets.

**Initialization**    The target of initialization is to provide convolutional kernels with a good starting point capturing class-specific sub-sequences in the data. As illustrated in the left part of Figure 2, the main idea is that for every class of data, we split them along the time axis, and calculate the mean of sub-sequences in each split part. Suppose we have $n_{out}$ output channels, corresponding to $n_{out}$ shapelets, with each shapelet of length $l_s$. The input length is $l_X$, and there are $n_{cla}$ classes. First, we assign $k = n_{out}/n_{cla}$ shapelets to each class. Within each class, we divide the time series into $k$ equal parts of length $l_k = l_X/k$ (along the time axis). For each part, we use a sliding window to find all possible candidates of length $l_s$. The final initialization of the kernel is obtained by calculating the mean of all candidates. This method not only incorporates class information into the kernel but also associates each kernel with a specific region of the time-series data in the dataset. As a result, the model can more effectively capture local information within the initialization region. The initialization is also paralleled on GPU, making it very time efficient.

**Classifier and Loss Function**    The output of the ShapeConv layer is the shapelet transformed distances, representing features extracted by learned shapelets, and these features are further utilized by the downstream classifier. Here, we append a multi-layer perceptron (MLP) after the ShapeConv layer, to map the shapelet transformed distance to the class labels. This method allows for end-to-end training of the entire model, optimizing both the ShapeConv layer and the classifier simultaneously. The loss function is designed as follows,

$$\mathcal{L} = \mathcal{L}_{cls} + \lambda_{shape}\mathcal{R}_{shape} + \lambda_{div}\mathcal{R}_{div}. \tag{8}$$

Here, $\mathcal{L}_{cls}$ is the task-specific classification loss, such as cross-entropy. The term $\mathcal{R}_{shape}$ is the shape regularizer defined in Eq. 6, and the term $\mathcal{R}_{div}$ is the diversity regularizer defined in Eq. 7. The hyperparameter $\lambda_{shape}$ and $\lambda_{div}$ controls the balance between each terms. Note that ShapeConv is compatible with other classifiers. We discuss this in Appendix D.1 and include this variant in our experiment in Sec. 4.1.

### 3.4 Unsupervised Learning

We now apply ShapeConv to the unsupervised learning task, where ShapeConv needs to capture the most representative sub-sequences in data and perform K-means on the shapelet-transformed distance to cluster unlabelled time series.

**Initialization** Initialization in the unsupervised learning setting is more challenging, since no class information is given now. Therefore, we design a pre-clustering method to find out some discriminative sub-sequences. The main idea is dividing the time-series data into portions along the time axis and performing separate clustering within each portion to find initial shapelets, as illustrated in the right part of Figure 2. Suppose we have $n_{out}$ shapelets in the ShapeConv layer, each with a length of $l_s$. The input length is $l_X$. First, all input time series are divided into $n_{cut}$ equal parts along the time axis, each with a length of $l_{cut} = l_X/n_{cut}$. Each part is assigned with $k = n_{out}/n_{cut}$ shapelets. We then sample a large number of subsequences (e.g., 10,000) of length $l_s$ from each part and perform KMeans clustering with $k$ centers on them. Finally, the cluster centers are utilized as the initialization of the shapelets. In this approach, the class information is implicitly introduced during the clustering process, and the division into cuts allows the shapelets to focus on different regions of the time series. This enables the model to better capture the local patterns of different classes.

**Loss Function** As for the task-specific loss, we employ Davies-Bouldin Index (DBI) (Davies & Bouldin, 1979) to optimize ShapeConv for better clustering results. Overall, DBI loss aims to minimize the intra-cluster distances while maximizing the inter-cluster distances, ensuring that the extracted shapelet transformed distances yield well-separated clusters (Li et al., 2022). The detailed formulation of DBI loss can be found in Appendix D.2. The overall loss function in unsupervised learning is designed as,

$$\mathcal{L} = \mathcal{L}_{\mathrm{DBI}} + \lambda_{shape}\mathcal{R}_{shape} + \lambda_{div}\mathcal{R}_{div}. \tag{9}$$

Here, $\mathcal{L}_{DBI}$ is DBI loss. The term $\mathcal{R}_{shape}$ is the shape regularizer defined in Eq. 6, and the term $\mathcal{R}_{div}$ is the diversity regularizer defined in Eq. 7. The hyperparameter $\lambda_{shape}$ and $\lambda_{div}$ controls the balance between each terms.

**Incorporating Human Knowledge** The characteristic of ShapeConv makes it easy to incorporate human knowledge, which means that human experts can "tell" the model what some key sub-sequences look like, and the model can use these knowledge for improving its performance. On the other hand, in unsupervised learning tasks, the model will first learn to minimize the shapelet transform distance, which may lead it to converge to local minima. If the shapelet is initialized in a non-discriminative region, the model's performance may be negatively affected. Therefore, we propose using human knowledge for shapelet initialization.

Specifically, we first visualize the dataset and ask the human labeler to identify the most discriminative regions. Once these regions are labeled, we calculate the mean of each region and use it as the initialization for the shapelets. Then, the shapelet will tend to converge in the targeted region. As shown in experiments in Section 4.2, this approach makes the model learn high-quality shapelets.

## 4 Experiments and Analysis

### 4.1 Supervised Time-Series Classification

**Settings** We evaluate our ShapeConv model on time-series classification tasks using the UCR univariate time-series dataset (Dau et al., 2019) and UAE multivariate times series dataset (Bagnall et al., 2018). Hyperparameters are tuned via grid search based on the validation set performance, and they are reported in Appendix G.2.

**Compared Methods** We compare ShapeConv with three kinds of baselines: (1) shapelet-based methods (IGSVM (Hills et al., 2014), FLAG (Hou et al., 2016), LTS (Grabocka et al., 2014), ADSN (Ma et al., 2020b)), (2) common deep learning methods (MLP, CNN, ResNet (Wang et al., 2017)), and (3) state-of-the-art time-series classification models (DTW (Chen et al., 2013), TNC (Tonekaboni et al., 2021), TST (Zerveas et al., 2021), TS-TCC (Eldele et al., 2021), T-Loss (Franceschi et al., 2019), TS2Vec (Yue et al., 2022)). The results for the baseline methods are taken directly from the

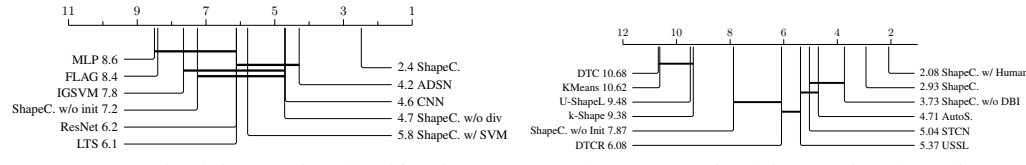

| (a) Supervised time-series classification | (b) Unsupervised time-series clustering |

Figure 3: Critical difference diagram for supervised and unsupervised learning tasks. Pairwise statistical comparison is based on accuracy on 25 datasets from the UCR Archive for supervised classification, and NMI on 36 datasets from the UCR Archive for unsupervised clustering.

Table 1: Testing accuracy of supervised time-series classification tasks on UCR and UEA datasets. Training Time includes initialization.

| | 125 UCR datasets | | | 30 UEA datasets | | |
|---|---|---|---|---|---|---|
| Method | Avg. Acc.[1] | Avg. Rank | Training Time (hours) | Avg. Acc. | Avg. Rank | Training Time (hours) |
| DTW | 0.727 | 6.10 | – | 0.650 | 4.72 | – |
| TNC | 0.761 | 5.29 | 228.4 | 0.670 | 4.58 | 91.2 |
| TST | 0.641 | 7.06 | 17.1 | 0.617 | 5.28 | 28.6 |
| TS-TCC | 0.757 | 5.19 | 1.1 | 0.668 | 4.33 | 3.6 |
| T-Loss | 0.806 | 4.52 | 38.0 | 0.658 | 3.90 | 15.1 |
| TS2Vec | 0.836 | 2.86 | 0.9 | 0.704 | 3.12 | **0.6** |
| ROCKET | 0.842 | 2.59 | 0.89 | – | – | – |
| ShapeConv | **0.851** | **2.39** | **0.5** | **0.750** | **2.07** | 0.8 |

original paper. In addition, we also design some variants of ShapeConv for ablation studies, including training without diversity loss (w/o div), training with random initialization (w/o init), and using an SVM classifier (w/ SVM).

Table 2: Testing accuracy and training time of ShapeConv and LTS on the Herring dataset.

| Shapelet Length | | 50 | 100 | 150 | 200 | 250 | 300 | 350 | 400 |
|---|---|---|---|---|---|---|---|---|---|
| Accuracy | ShapeConv | 61.3 | 67.2 | 67.2 | 68.8 | 75.0 | 70.3 | 71.9 | 71.9 |
| | LTS | 64.1 | 60.9 | 59.4 | 59.4 | 59.4 | 59.4 | 59.4 | 59.4 |
| Time | ShapeConv | 9.20s | 9.09s | 9.15s | 9.30s | 9.27s | 9.19s | 9.02s | 9.09s |
| | LTS | 3.37h | 3.20h | 3.23h | 3.17h | 2.69h | 2.25h | 1.87h | 1.21h |

**Results** The performance of ShapeConv, its variants, and shapelet-based baselines, is evaluated on the 25 UCR datasets and presented in Figure 3 (a). For this experiment, we compared in the 25 subset of the 128 UCR dataset, because the baseline method only reported on this subset. Additionally, the summary of results for state-of-the-art time-series classification models on 125 UCR and 29 UEA datasets are shown in Table 1. The full results are presented in the Appendix G.3 In general, ShapeConv consistently outperforms all other baselines and variants, ranking first on average. These results demonstrate that ShapeConv not only provides interpretability but also excels in performance, making it a competitive choice for time-series classification compared to state-of-the-art methods. Ablation studies tell that the effectiveness of our initialization method and diversity loss contributes to improved performance compared to the variants. Lastly, the choice of downstream classifier, either SVM or MLP, does not significantly impact the performance of the ShapeConv model, indicating its flexibility and robustness in different classification settings.

**Analysis** In this section, we investigate two main research questions (RQs): (1) why does ShapeConv outperform other shapelet-based methods? (2) how are ShapeConv's interpretability results compared to other shapelet-based methods when they yield similar results?

In response to the first RQ, we examine the Herring datasets from UCR (Dau et al., 2019). Learned shapelets with minimum distance from the original data by the model with best validation accuracy are

---

[1]This average accuracy metric is meaningless to some extent, due to datasets of different sizes, class skews, number of classes, default rates, etc. We list these results here just for the comparison with previous works, but we sincerely call for metrics with more practical values here.

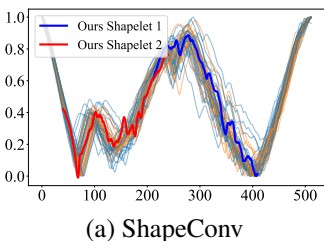 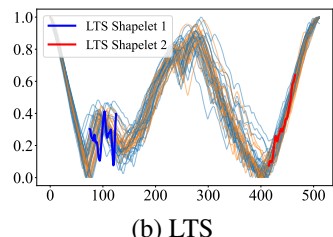 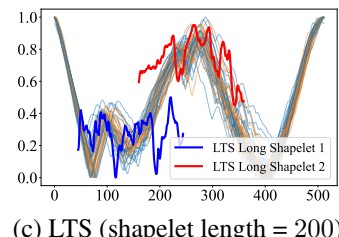

(a) ShapeConv          (b) LTS          (c) LTS (shapelet length = 200)

Figure 4: Learned shapelets by different methods on the Herring dataset.

plotted in Figure 4. In the Herring dataset, ShapeConv (test accuracy 75.0) significantly outperforms the LTS method (test accuracy 64.1). In response to the first RQ, we observe that ShapeConv's shapelets (Figure 4 (a)) cover the most discriminative regions of the time series (the turning points), while LTS's shapelets (Figure 4 (b)) do not. This indicates that ShapeConv's learned shapelets are better at distinguishing classes, leading to improved performance.

We find that the learned shapelet by ShapeConv is much longer than that by LTS. The result of forcing the shapelet learned by LTS to be longer (Figure 4 (c)) reveals that LTS fails to learn a high-quality long shapelet. We also provide an ablation on the shapelet length in Table 2. It shows ShapeConv's accuracy increases with shapelet length up to a certain point, while LTS's accuracy does not benefit from the increased length. This is likely due to optimization issues in the LTS method, which cannot handle long-length shapelets. ShapeConv, on the other hand, can be efficiently computed in parallel, leading to better optimization results and significantly faster training time (about 1000× faster).

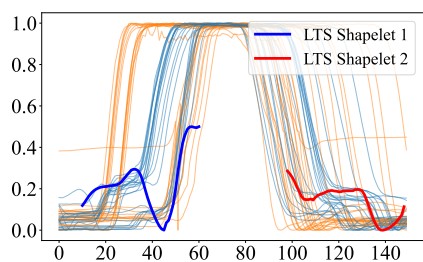

Figure 5: Shapelets learned by LTS in the GunPoint dataset.

In the GunPoint dataset, both ShapeConv and LTS methods achieve saturated accuracy (100). In response to the second RQ, we utilize this dataset to compare the interpretability of the learned shapelets by ShapeConv (Figure 1) and LTS (Figure 5). It is evident that the shapelet learned by ShapeConv captures the distinguishing features of the class effectively. Here, the shapelet 1 (blue) captures the gesture of reaching for the gun and drawing it out of the holster. The shapelet 2 (red) captures the gesture of putting the gun back to the holster. In contrast, the shapelets learned by LTS do not align well with either of the classes, especially for shapelet 1 in blue. Based on this observation, we conclude that ShapeConv is capable of learning more interpretable shapelets compared to LTS.

## 4.2 UNSUPERVISED TIME-SERIES CLUSTERING

**Settings** We evaluate our ShapeConv model on time-series clustering task using 36 UCR univariate time-series datasets (Dau et al., 2019). We first learn shapelets using a ShapeConv layer, then apply KMeans on the shapelet-transformed distance. We use the Normalized Mutual Information (NMI) metric to evaluate the models. Hyperparameters are tuned via grid search based on validation set performance, and they are reported in Appendix G.2.

**Compared Methods** We compare ShapeConv with three kinds of baselines: (1) pure clustering methods (KMeans (Hartigan & Wong, 1979) applied to the entire time series), (2) shapelet-based methods (U-Shapelet (Zakaria et al., 2012), AutoShape (Li et al., 2022)), and (3) state-of-the-art time-series clustering models (k-Shape (Paparrizos & Gravano, 2015), DTC (Madiraju et al., 2018), USSL (Zhang et al., 2018), DTCR (Ma et al., 2019), STCN (Ma et al., 2020a)). The results for the baseline methods are taken directly from the original paper. We also design variants of ShapeConv for ablation studies, including training with random initialization (w/o Init), training without DBI Loss (w/o DBI), and using human knowledge to initialize shapelets (w/ Human).

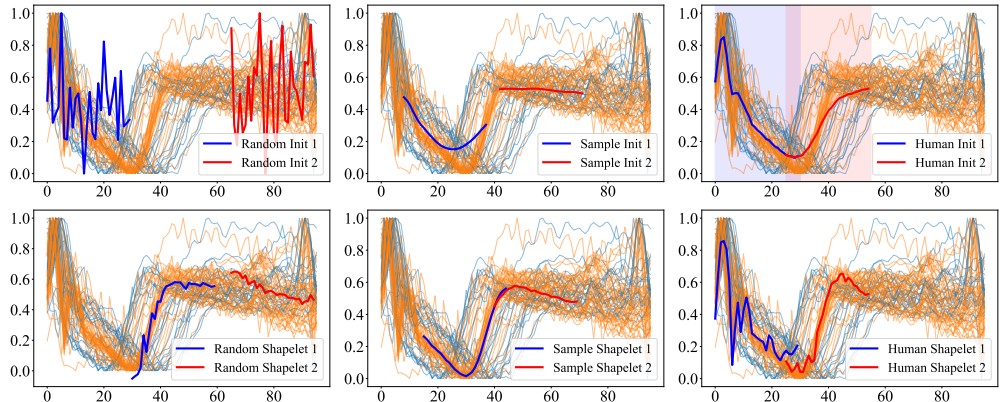

Figure 6: Visualization of cases from ECG200 dataset. First row: illustration of the shapelet initializations; second row: illustration of the learned shapelets. Left: random initialization; middle: cut and sample initialization (Sec. 3.4); right: human knowledge initialization (Sec. 3.4)

**Results** The results of all models on 36 UCR datasets are shown in Figure 3 (b), and details are in Appendix G.4. We compared on the 36 subset of the UCR datasets because the baseline methods only reported on this subset. In general, our ShapeConv outperforms all other baselines and variants, achieving the highest average rank among the compared methods. ShapeConv's superior performance, particularly against its randomly initialized variant, underscores the importance of proper initialization for effective clustering. Its best performance with human knowledge initialization highlights the model's ability to incorporate human knowledge to guide the learning process and improve clustering results. Overall, ShapeConv stands out as a potent, interpretable tool for unsupervised time-series clustering, outperforming existing methods while adeptly learning interpretable shapelets and assimilating human knowledge.

**Analysis of the Initialization** We now provide a case study to analyze the effect of initialization for ShapeConv in unsupervised learning tasks. We select the ECG200 dataset from UCR (Dau et al., 2019) for this analysis and results are plotted in Figure 6. First, we observe that in the time-series clustering task, *the learned shapelets are close to their initializations*. This is because, during the first step of learning shapelets, we solely minimize the shapelet-transformed distance, which tends to optimize within the local region. Therefore, determining the initialization of the shapelets is critical for unsupervised learning.

In both the random and sample initialization, one of the shapelets matches the right part of the time series, where the two classes are indistinguishable. In contrast, when using human initialization, we choose the two regions with the most significant differences between the classes (the shaded regions in Figure 6) and use the average of those regions as initialization. Consequently, the shapelets are converged in the these regions, effectively capturing the differences between the classes.

## 5    CONCLUSION

In this paper, we bring together CNNs and shapelets in time-series modeling by finding the equivalance between them. Upon the findings, we further proposed ShapeConv, an interpretable convolutional kernel with its kernels serving as shapelets accompanied by shaping regularizations, and we apply ShapeConv to both supervised and unsupervised tasks. ShapeConv is designed to maintain the advantages of both CNNs and shapelets, providing excellent performance without sacrificing interpretability and controllability. Our experiments on various benchmark datasets showed that ShapeConv outperforms other shapelet-based methods and state-of-the-art time-series classification and clustering models. Moreover, the incorporation of human knowledge can further enhance the performance of ShapeConv, highlighting its potential in real-world applications where expert knowledge is available.

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

APPENDIX

## A    RELATED WORK

**Learning Shapelets** How to find the best shapelets from data has long been an intriguing problem since (Ye & Keogh, 2009) firstly proposed it. Traditional practice is to *search the raw datasets* with some speed-up strategies, like paralleling computing (Chang et al., 2012), SAX transformation (Rakthanmanon & Keogh, 2013) and procedure simplification through newly designed measurements (Lines et al., 2012; Guillaume et al., 2022; Zakaria et al., 2012). However, despite ingenious techniques, the performance of these methods are quite limited in large-scale real-world scenarios due to their inefficiency and inflexibility. Cutting-edge research mostly focus on *learning shapelets via optimization-based methods*. (Grabocka et al., 2014) firstly proposed to learn shapelets with gradient descent. (Shah et al., 2016; Lods et al., 2017) extended this idea to learn more discriminative shapelets based on DTW measure. To encourage the interpretability of learned shapelets, (Ma et al., 2020b; Wang et al., 2019) designed adversarial strategies to guide model training. Besides, in (Li et al., 2022; Zhang & Sun, 2022), autoencoder and neighbour graph structure were also leveraged to capture high-quality shapelets in an unsupervised manner. Nevertheless, none of these methods have rigorously shown and made full use of the equivalence between the CNN layer and shapelet to achieve both good interpretability and efficiency like ours.

**Interpretable Time Series Modeling** Despite noticeable progress made by feature engineering methods (Ruiz et al., 2021; Bagnall et al., 2017; Middlehurst et al., 2023) like HIVE-COTE (Lines et al., 2018), MUSE (Schäfer & Leser, 2017) and ROCKET (Dempster et al., 2020), recent years have also witnessed the rapid advancement of deep learning methods in interpretable time series modeling. Apart from RNN models (Choi et al., 2016; Guo et al., 2019), newly devised CNN models have obtained more attention in this area. (Fortuin et al., 2018) developed a CNN-based SOM-VAE method to learn the topologically interpretable discrete representations of time series in a probabilistic fashion. (Luo et al., 2022) employed convolutional kernels to approximate the partial differential equations on data distribution so as to explain the nonlinear dynamics of their sequential patterns. (Li et al., 2021) designed a wavelet convolution layer to help CNNs discover filters with certain physical meaning, while (Tang et al., 2021b; Xiao et al., 2022) studied the best kernel size for time series modelling. Innovatively, we view convolution kernels from the shapelet perspective and endow shapelet-based interpretability to incomprehensible model parameters, making ShapeConv an interpretable model.

## B    DETAILED EXPLANATIONS ON SHAPELETS AND SHAPELET TRANSFORM

### B.1    INTERPRETABILITY AND EXPAINABILITY

The interpretability of shapelet comes from its human-comprehensible nature. However, "interpretability" is sometimes confounded with the concept of "explainability", introduced in many post-hoc explainable methods (Tonekaboni et al., 2020; Leung et al., 2021). These explainable methods focus on explaining an already learned, non-interpretable model. They may not reflect the actual behavior of the original model and may disagree with each other (Krishna et al., 2022). Self-interpretable methods such as shapelets do not have these issues as the model itself provides explanations during the learning process. Furthermore, the proposed ShapeConv in our paper whose kernel serves as shapelet inherit the same interpretability of the traditional shapelet, utilizing the prototypical shape information of the data to perform classification.

### B.2    VISUALIZING SHAPELETS AND SHAPELET TRANSFORM BY SHAPECONV

To illustrates shapelets and shapelet transform more vividly, we take a real-dataset example named *GunPoint*, aiming at classifying whether a person is pointing with a finger or a gun (Figure B.1). The sequences in the dataset record the normalized x-axis of the hand position, i.e., how far away the hand is from the main body through time.

Figure B.2 demonstrates the result. The leftmost two subfigures show the learned shapelet. When we try to align the shapelets with typical samples one with "gun" (Class 1) in blue and one with "finger" (Class 2) in orange in the middle subfigure, we can immediately find that the most distinguishable shapes underlying data from Class 1 are the small flat stage soon after the beginning and symmetrically

in the end. These two shapes corresponds to the motion of a hand pulling a gun out of the holster and put it back, and would not exist when a person is pointing with his finger.

Therefore, when we perform the shapelet transform by calculating the distance between data points and the learned shapelet and draw them in the 2-D feature plane as is shown in the right subfigure, we found the two classes are linear separable as expected. Data points at the left-bottom of the plane from the "gun" class contain the shapelets indicating the gun-related actions, while data from the "finger" class do not. This illustrate how the model can be discriminative while providing interpretability at the same time.

In order to further verify our interpretations, we obtained the frames of videos during the data collection process from the collector and visualized the trajectory of hand movements for the first half of the sequence, as is shown in Figure B.3. For data of the *Gun* class, we observe clear stops when pulling the gun out of the holster as is highlighted in green circles, and such movements do not exist in the data of the *Point* class. This observation matches our hypothesis made upon the learned shapelets.

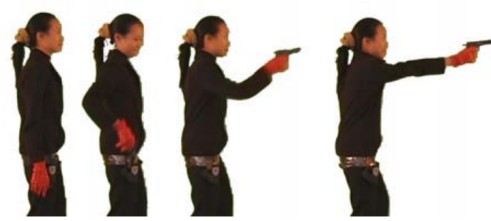

Figure B.1: Illustration of the GunPoint Dataset (Ratanamahatana & Keogh, 2005) [2]

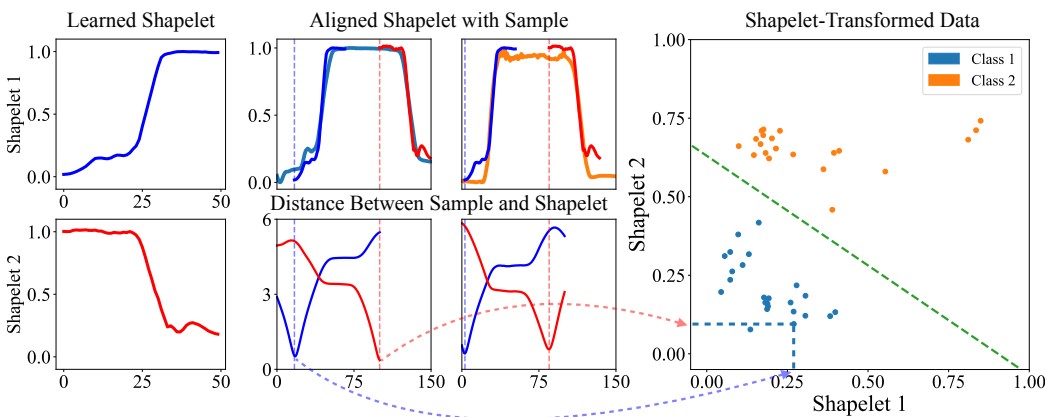

Figure B.2: Illustration of shapelets and the shapelet transform step

## C    PROOF OF THEOREM 3.1

**Theorem 3.1** *Assume the input* $\mathbf{X} \in \mathbb{R}^{l_x}$ *is a 1-dimensional single-variate signal of length* $l_x$, *and* $n_{out}$ *shapelets* $S^* = \{\mathbf{s_1}, \mathbf{s_2}, ... \mathbf{s_{n_{out}}}\}$ *with length* $l_s$ *are discovered. The feature extracted from* $\mathbf{X}$ *with* $\mathbf{s_i}$ *and squared Euclidean distance is* $d_{\mathbf{s_i}, \mathbf{X}}$. *Then we have*

$$d_{\mathbf{s_i}, \mathbf{X}} = -2 \max_{j \in \{1, 2, ..., l_x - l_s + 1\}} [\mathbf{Y}_{ij} - \mathcal{N}(\mathbf{s_i}, \mathbf{X}_{j:j+l_s-1})], \quad (10)$$

*where* $\mathbf{Y} = \mathbf{s_i} * \mathbf{X}$ *is the cross-correlation defined in Eq. 2 and* $\mathcal{N}(\mathbf{s_i}, \mathbf{X}') = (\|\mathbf{s_i}\|_2^2 + \|\mathbf{X}'\|_2^2)/2$ *is squared* $L_2$ *norm term.*

---

[2]Image    Source:    `http://www.timeseriesclassification.com/description.php?Dataset=GunPoint`

Gun

Point

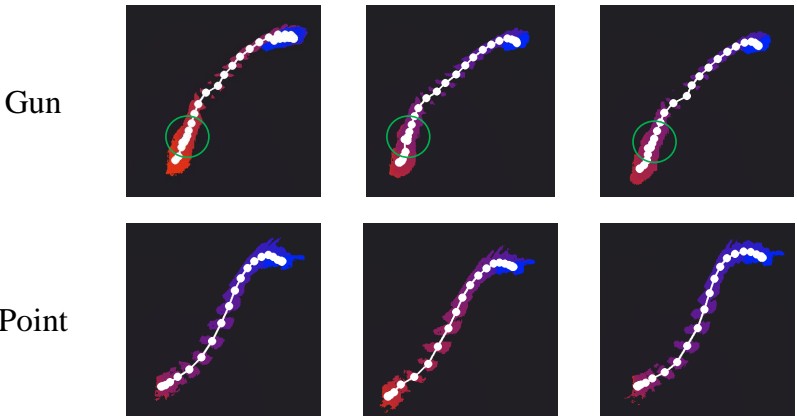

Figure B.3: Illustration of the hand movements in GunPoint dataset

*Proof.* According to Eq.1, shapelet transform step extract features using minimal distance and can be expanded as:

$$d_{\mathbf{s_i}, \mathbf{X}} = \min_{\mathbf{x} \in \hat{\mathbf{X}}} dist(\mathbf{s_i}, \mathbf{x}) = \min_{j \in \{1, 2, \cdots, l_x - l_s + 1\}} \| \mathbf{s_i} - \mathbf{X}_{j:j+l_s-1} \|_2^2$$

$$= \min_{j \in \{1, 2, \cdots, l_x - l_s + 1\}} \left( \| \mathbf{s_i} \|_2^2 + \| \mathbf{X}_{j:j+l_s-1} \|_2^2 - 2 \sum_{k=1}^{l_s} \mathbf{s_{i}}_k \mathbf{X}_{j+k} \right)$$

$$= \min_{j \in \{1, 2, \cdots, l_x - l_s + 1\}} \left[ \| \mathbf{s_i} \|_2^2 + \| \mathbf{X}_{j:j+l_s-1} \|_2^2 - 2(\mathbf{s_i} * \mathbf{X})_k \right],$$

$$= -2 \max_{j \in \{1, 2, \cdots, l_x - l_s + 1\}} \left[ \mathbf{Y}_{ij} - \mathcal{N}(\mathbf{s_i}, \mathbf{X}_{j:j+l_s-1}) \right]$$

with $\mathbf{Y}$ and $\mathcal{N}(\mathbf{s_i}, \mathbf{X}_{j:j+l_s-1})$ defined as in the theorem. $\square$

## D  DETAILS ON MODEL DESIGNS

### D.1  COMPATIBLE WITH OTHER CLASSIFIERS

ShapeConv can also be used together with traditional classifiers, such as support vector machines (SVMs), decision trees, or random forests. In this case, the whole model cannot be optimized via an end-to-end fashion, so we decompose the shapelet learning step and classification. Specifically, no additional module is appended to the ShapeConv layer, and the output of the layer is directly optimized using the above loss function, but with $\mathcal{L}_{cls}$ term in Eq. 8 removed. Then, the learned features are fed into the chosen classifier for training and prediction. We also include this variant in our experiment in Sec. 4.1.

### D.2  DAVIES-BOULDIN INDEX (DBI) LOSS

When the number of cluster is set to $k$, DBI can be denoted as

$$\mathcal{I}_{\text{DBI}} = \frac{1}{k} \sum_{i=1}^{k} \max_{j=1 \ldots k, j \neq i} \frac{r_i + r_j}{d_{ij}}. \tag{11}$$

Here, $r_i$ is the diameter of cluster $i$, which is defined as the average distance between each element in cluster $i$ and the center of cluster $i$. The distance between the center of cluster $i$ and cluster $j$ is $d_{i,j}$. However, this formulation is not tractable for optimization due to the max operator. Thus, following Li et al. (2022), the max operator is replaced and approximated by the following calculation,

$$\mathcal{L}_{\text{DBI}} = \frac{1}{k} \sum_{i=1}^{k} \frac{\sum_{j=1}^{k} m_{ij} \cdot e^{\alpha m_{ij}}}{\sum_{j=1}^{k} e^{\alpha m_{ij}}}, \tag{12}$$

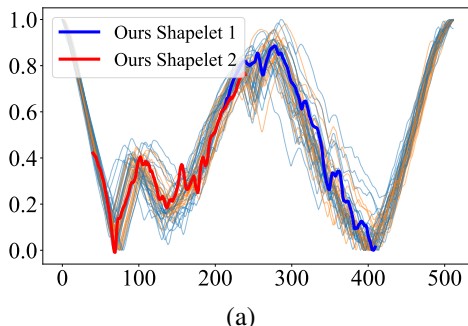 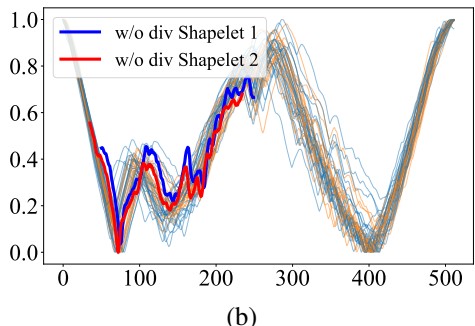

|   (a)   |   (b)   |

Figure D.1: Learned shapelets by different methods on the Herring dataset. (a) ShapeConv; (b) ShapeConv without diversity loss.

where $m_{ij} = \frac{r_i + r_j}{d_{ij}}$. By numerical verification, $\alpha = 100$ is enough for Eq. 12 to approximate the true maximum.

### D.3   ILLUSTRATION OF DIVERSITY LOSS

Figure D.1 depicts the trained shapelet both with and without diversity loss. The illustration reveals that without the application of diversity regularization, the shapelets tend to converge to the same local minimum. Employing diversity loss can mitigate this issue.

## E   SHAPECONV AS FEATURE EXTRACTOR WITH DEEP MODELS

Another advantage brought by our ShapeConv to find shapelets with a special kind of convolutional layer is the flexibility. While the traditional shapelet works suffer from handling large time-series data of long sequence efficiently, ShapeConv turns the extraction of shapelets from the original data into a stackable layer can be combined with more sophisticated deep models and optimized in an end-to-end manner, leaving the possibility to keep the interpretability and effectiveness to the maximum extent.

To verify the effectiveness of ShapeConv embedded in a deep model, we apply ShapeConv as the first layer to extract features which are further processed by GRU(Cho et al., 2014), a widely used time-series model for long-term modeling. We then conducted the experiment of the proposed method on the seizure detection task based on electroencephalograph (EEG) data (Obeid & Picone, 2016). The dataset contains 97,859 samples (83,647 for training and 14,212 for testing), and each sample contains 20-channel 30-second EEG signal sampled at 200Hz. The goal of the prediction models is to predict the probability of seizure event within the given EEG signal piece, following (Tang et al., 2021a; Li et al., 2023).

We compare our model ShapeConv with the most commonly used deep neural network models GRU (Cho et al., 2014) and TCN (Bai et al., 2018). The empirical results are illustrated in Table E.1. Our model has significantly outperformed the compared baselines, which showed the superiority of the proposed ShapeConv paradigm even embedded in another neural architectures.

Table E.1: Performance comparison on seizure detection task.

| Model | AUROC | AUPRC |
|---|---|---|
| GRU Cho et al. (2014) | 0.814(0.009) | 0.386(0.018) |
| TCN Bai et al. (2018) | 0.817(0.004) | 0.383(0.010) |
| ShapeConv + GRU | **0.837(0.007)** | **0.414(0.008)** |

We further investigate whether our ShapeConv can preserve its interpretability when stacking with deep models. As clinical practice, the morphology of waveform in EEG describes its overall shape, and is important for both interpreting a tracing and communicating findings, which has been well-studied and recognized in previous medical research (Marcuse et al., 2015). To this end, we visualized

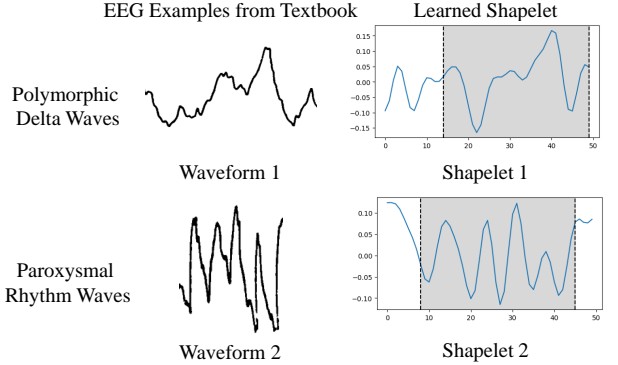 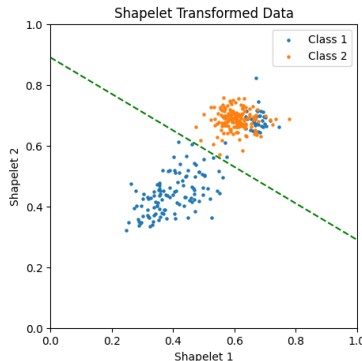

(a) Comparison between two types of textbook waveform and the two learned shapelets, polymorphic delta waves related to lesions (Marshall et al., 1988) and paroxysmal rhythm waves related to sleep stage (Alvarez et al., 1983).

(b) Shapelet transform by the two shapelets. Some data points from Class 1 are mixed with Class 2 need to be separated by other shapelets.

Figure E.1: Demonstration of the learned shapelets for EEG data.

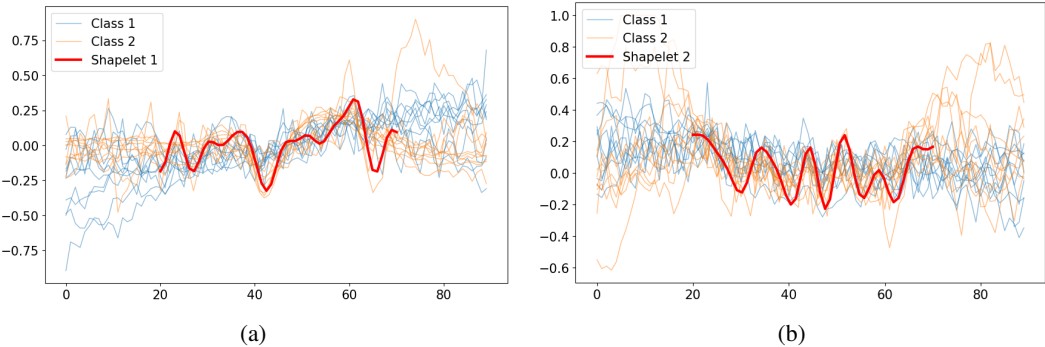

(a)

(b)

Figure E.2: Alignment between the learned shapelets and original data.

a few obtained shapelets out of 2,688 shapelets (128 shapelets per variate, 21 variates in total) and excitedly found that some of them accord with some textbook waveform, as is shown in Figure E.1a. By aligning them with the mostly similar part of the original data (Figure E.2), we found these shapelets can match a specific type of seizure status and provides solid classification criteria (Figure E.1b). This showcases the possibility of how our interpretable method can benefit medical practitioners in practice by not only offering an accurate judgement, but also pointing out the area of interests with respect to their expert knowledge.

It's also noteworthy that since the amount of summarized waveform in textbook is limited, some shapelets, while serving as similarly strong indicators, may not be included in existing studies. We believe these shapelets can provide inspirations and boost further research in related area.

## F    MORE VISUALIZATIONS OF SHAPECONV

### F.1    MORE VISUALIZATIONS ON LEARNT SHAPELETS

In this section, we provide more visualizations of the learnt shapelets of ShapeConv in different UCR datasets. The results clearly shows that ShapeConv could learn the determining regions of the time series.

### F.2    VISUALIZATIONS WITH SHAP VALUE

In this section, we further substantiate our claim regarding the interpretability of our model using the SHAP (SHapley Additive exPlanations) Value (Lundberg & Lee, 2017). The analysis employs the GunPoint dataset from the UCR archive. We examine two variations of our model: the original

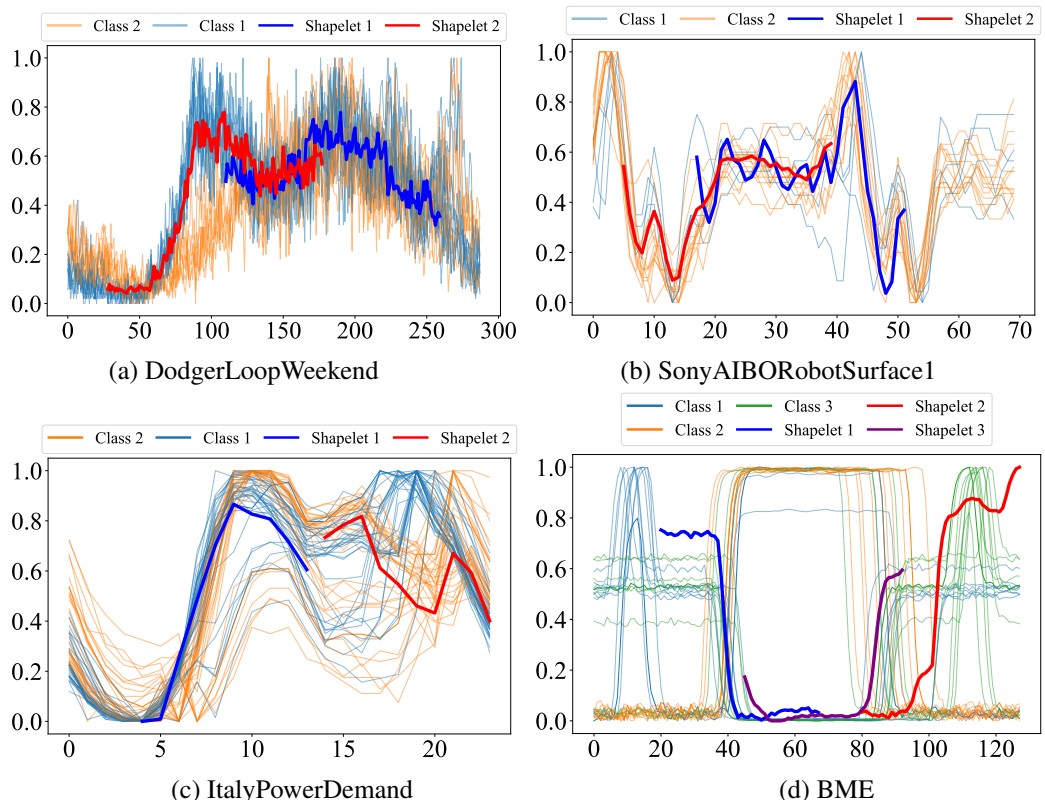

Figure F.1: More visualizations of the shapelets learnt by ShapeConv in UCR datasets.

ShapeConv and a modified version where the term $\lambda_{shape}$ in the loss function (Equation 8) is set to zero. Setting $\lambda_{shape}$ to zero eliminates the L2 norm term, effectively transforming the layer into a standard CNN. Consequently, this variant lacks the interpretability feature.

Both models underwent training under identical hyperparameters. Post-training, we computed the SHAP Values for each model across the entire test dataset using the expected gradients approach. These values are illustrated in Figure F.2, with the mean SHAP value of each class depicted. The blue and orange lines represent the Gun Class (Class 1) and No Gun Class (Class 2), respectively.

The left side of Figure F.2 reveals that the model is particularly sensitive to the left and right turning points. These points symbolize the gesture of drawing the gun out of the holster and putting it back, underscoring the model's reliance on these regions for decision-making. This observation aligns with our hypothesis about the model's interpretative capabilities. However, on the right, we first notice that the kernel does not match with the input sequence, indicating the lack of interpretability. Additionally, the model appears to base its decisions predominantly on the left region. This disparity highlights the limitations of the variant without the interpretability term.

In conclusion, our findings are twofold: firstly, the integration of Shape Loss successfully enhances interpretability. Secondly, ShapeConv not only encompasses all significant regions identified by the SHAP Value of the baseline CNN, but also surpasses conventional explainability methods like SHAP by capturing the shape of sensitive regions, rather than merely indicating their locations.

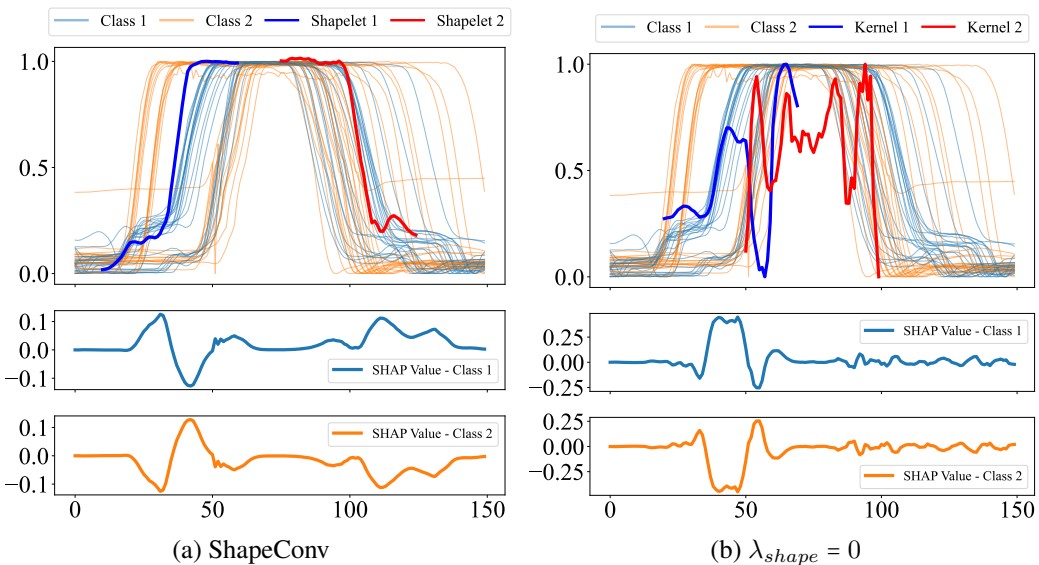

Figure F.2: Illustration of the trained average SHAP value for different class across all the GunPoint testing dataset. Blue: Gun Class. Orange: No Gun Class. Left: our proposed ShapeConv method. Right: ShapeConv with $\lambda_{shape} = 0$.

## G DETAILS ON EXPERIMENTS AND ANALYSIS

### G.1 ENVIRONMENT

All experiments are performed on the PyTorch framework using a 24-cores AMD Epyc 7V13 2.5GHz CPU, 220GB RAM, and an NVIDIA A100 80GB PCIe GPU. The server is provided by the Azure cloud computing platform.

### G.2 HYPERPARAMETERS

**Supervised Learning** The training set is divided into training and validation sets at an 8:2 ratio. Hyperparameters are tuned via grid search based on validation set performance. The number of shapelets is chosen from $\{1, 2, 3, 4, 5\}$ times the number of classes, and the shapelet length is evaluated over $\{0.1, 0.2, \cdots, 0.8\}$ times the time series length. The parameter $\lambda_{shape}$ is chosen from $\{0.01, 0.1, 1, 10\}$ and the parameter $\lambda_{div}$ is evaluated over $\{0.01, 0.1, 1, 10\}$. Learning rate is chosen from $\{0.001, 0.005, 0.01, 0.05, 0.1\}$.

**Unsupervised Learning** The training set is divided into training and validation sets at an 8:2 ratio. Hyperparameters are tuned via grid search based on validation set performance. The number of shapelets is chosen from $\{1, 2, 3, 4, 5\}$ times the number of classes, and the shapelet length is evaluated over $\{0.1, 0.15, 0.2, 0.25, \cdots, 0.8\}$ times the time series length. The parameter $\lambda_{shape}$ is chosen from $\{0.01, 0.1, 1, 10\}$ and the parameter $\lambda_{div}$ is evaluated over $\{0.01, 0.1, 1, 10\}$. Learning rate is chosen from $\{0.001, 0.005, 0.01, 0.05, 0.1\}$.

### G.3 RESULTS OF SUPERVISED LEARNING TASKS

In this section, we present the full results of supervised time-series classification tasks. We compared ShapeConv with (1) shapelet-based methods, common deep learning methods, and ablations (described in Sec. 4.1) across 25 UCR Datasets (Table G.1) (2) RNN-based methods (Tang et al., 2021b) across 56 UCR Datasets (Table G.2) (3) state-of-the-art times series classification methods (described in Sec. 4.1) across 125 UCR datasets (Table G.3) (4) state-of-the-art times series classification methods (described in §4.1) across 30 UEA datasets (Table G.4).

Table G.1: ShapeConv Compared with Shapelet-Based Methods, Common Deep Learning Methods, and Ablations: Evaluating Testing Accuracy for Supervised Time-Series Classification Tasks Across 25 UCR Datasets. Mean accuracy ± std over 3 independent experiments with different random seeds is reported.

| Dataset | MLP | CNN | ResNet | IGSVM | FLAG | LTS | ADSN | ShapeC. w/o init | ShapeC. w/ SVM | ShapeC. w/o div | ShapeC. |
|---|---|---|---|---|---|---|---|---|---|---|---|
| Adiac | 0.752 | 0.857 | 0.826 | 0.235 | 0.752 | 0.519 | 0.798 | 0.691±0.007 | 0.813±0.047 | 0.852±0.014 | **0.867**±0.020 |
| Beef | 0.833 | 0.750 | 0.767 | 0.900 | 0.833 | 0.767 | 0.933 | 0.849±0.005 | 0.898±0.031 | 0.921±0.001 | **0.936**±0.008 |
| Chlorine. | 0.872 | 0.843 | 0.828 | 0.571 | 0.760 | 0.730 | 0.880 | 0.825±0.026 | 0.904±0.025 | 0.907±0.008 | **0.924**±0.014 |
| Coffee | **1.000** | **1.000** | **1.000** | **1.000** | **1.000** | **1.000** | **1.000** | **1.000**±0.000 | **1.000**±0.000 | **1.000**±0.000 | **1.000**±0.000 |
| Diatom. | 0.964 | 0.930 | 0.931 | 0.931 | 0.964 | 0.942 | 0.987 | 0.992±0.008 | 0.991±0.001 | 0.992±0.005 | **0.994**±0.009 |
| DPLittle | 0.701 | 0.703 | 0.701 | 0.666 | 0.683 | **0.734** | 0.727 | 0.698±0.012 | 0.706±0.003 | 0.703±0.000 | 0.713±0.013 |
| DPMiddle | 0.721 | 0.736 | 0.723 | 0.695 | 0.713 | 0.741 | 0.784 | 0.778±0.018 | 0.782±0.003 | 0.789±0.001 | **0.807**±0.039 |
| DPThumb | 0.705 | 0.701 | 0.705 | 0.696 | 0.705 | 0.752 | 0.736 | 0.729±0.010 | 0.715±0.037 | 0.738±0.005 | **0.753**±0.015 |
| ECGFiveDays | 0.970 | 0.985 | 0.955 | 0.990 | 0.920 | **1.000** | **1.000** | **1.000**±0.000 | **1.000**±0.000 | **1.000**±0.000 | **1.000**±0.000 |
| FaceFour | 0.830 | 0.932 | 0.932 | **0.977** | 0.909 | 0.943 | **0.977** | 0.941±0.023 | 0.916±0.001 | 0.942±0.015 | 0.961±0.007 |
| GunPoint | 0.933 | **1.000** | 0.993 | **1.000** | 0.967 | 0.996 | 0.987 | 0.994±0.016 | 0.995±0.008 | 0.993±0.003 | 0.997±0.016 |
| Herring | 0.641 | 0.681 | 0.641 | 0.641 | 0.641 | 0.641 | 0.703 | 0.703±0.014 | 0.713±0.005 | 0.702±0.006 | **0.724**±0.028 |
| ItalyPower. | 0.966 | 0.970 | 0.960 | 0.937 | 0.946 | 0.958 | 0.972 | 0.932±0.027 | 0.953±0.005 | 0.961±0.001 | **0.974**±0.007 |
| Lightning7 | 0.644 | **0.863** | 0.836 | 0.630 | 0.767 | 0.790 | 0.808 | 0.758±0.003 | 0.723±0.001 | 0.748±0.001 | 0.781±0.004 |
| MedicalImages | 0.729 | **0.792** | 0.772 | 0.552 | 0.714 | 0.713 | 0.720 | 0.682±0.012 | 0.694±0.000 | 0.752±0.009 | 0.774±0.003 |
| MoteStrain | 0.869 | **0.950** | 0.895 | 0.887 | 0.888 | 0.900 | 0.906 | 0.884±0.013 | 0.886±0.016 | 0.898±0.002 | 0.913±0.000 |
| MPLittle | 0.703 | **0.758** | 0.726 | 0.707 | 0.693 | 0.743 | **0.758** | 0.701±0.012 | 0.733±0.002 | 0.741±0.008 | 0.749±0.032 |
| MPMiddle | 0.750 | 0.800 | 0.775 | 0.769 | 0.750 | 0.775 | 0.791 | 0.736±0.020 | 0.759±0.011 | 0.779±0.013 | **0.807**±0.020 |
| PPLittle | 0.710 | 0.753 | **0.761** | 0.721 | 0.671 | 0.710 | 0.715 | 0.661±0.013 | 0.694±0.029 | 0.676±0.000 | 0.732±0.001 |
| PPMiddle | 0.707 | 0.784 | 0.753 | 0.759 | 0.738 | 0.749 | 0.786 | 0.717±0.023 | 0.726±0.010 | 0.764±0.002 | **0.791**±0.000 |
| PPThumb | 0.726 | 0.745 | 0.708 | **0.755** | 0.674 | 0.705 | 0.695 | 0.685±0.018 | 0.712±0.010 | 0.728±0.010 | 0.731±0.011 |
| Sony. | 0.727 | 0.968 | **0.985** | 0.927 | 0.929 | 0.910 | 0.915 | 0.901±0.026 | 0.914±0.007 | 0.903±0.002 | 0.926±0.010 |
| Symbols | 0.853 | 0.962 | 0.872 | 0.846 | 0.875 | 0.945 | 0.963 | 0.942±0.026 | 0.968±0.006 | 0.974±0.007 | **0.980**±0.015 |
| SyntheticC. | 0.950 | 0.990 | **1.000** | 0.873 | 0.997 | 0.973 | **1.000** | **1.000**±0.000 | **1.000**±0.000 | 0.997±0.029 | **1.000**±0.000 |
| Trace | 0.820 | **1.000** | **1.000** | 0.980 | 0.990 | **1.000** | **1.000** | **1.000**±0.000 | **1.000**±0.000 | **1.000**±0.000 | **1.000**±0.000 |
| TwoLeadECG | 0.853 | **1.000** | **1.000** | **1.000** | 0.990 | **1.000** | 0.986 | **1.000**±0.000 | **1.000**±0.000 | **1.000**±0.000 | **1.000**±0.000 |
| Avg. Acc | 80.5 | 86.4 | 84.8 | 79.4 | 82.6 | 83.2 | 86.6 | 83.8 | 85.4 | 86.4 | **87.8** |
| Avg. Rank | 8.6 | 4.7 | 6.1 | 7.8 | 8.4 | 6.2 | 4.2 | 7.2 | 5.8 | 4.6 | **2.4** |

Table G.2: ShapeConv Compared with RNN-Based Methods: Evaluating Testing Accuracy for Supervised Time-Series Classification Tasks Across 56 UCR Datasets. Mean accuracy ± std over 3 independent experiments with different random seeds is reported.

| Dataset | RNTK | NTK | RBF | POLY | Gaussian RNN | Identity RNN | GRU | OS-CNN | ShapeConv |
|---|---|---|---|---|---|---|---|---|---|
| Adiac | 0.766 | 0.719 | 0.734 | 0.778 | 0.514 | 0.169 | 0.606 | 0.835 | **0.882**±0.009 |
| Arrowhead | 0.806 | 0.834 | 0.806 | 0.749 | 0.480 | 0.560 | 0.377 | 0.838 | **0.915**±0.033 |
| Beef | 0.900 | 0.733 | 0.833 | 0.933 | 0.267 | 0.467 | 0.367 | 0.807 | **0.941**±0.001 |
| Car | 0.833 | 0.788 | 0.800 | 0.800 | 0.233 | 0.583 | 0.267 | 0.933 | **0.992**±0.008 |
| ChlorineConcentration | 0.908 | 0.773 | 0.864 | 0.915 | 0.660 | 0.558 | 0.611 | 0.839 | **0.924**±0.002 |
| Coffee | **1.000** | **1.000** | 0.929 | 0.929 | **1.000** | 0.429 | 0.571 | **1.000** | **1.000**±0.000 |
| Computers | 0.592 | 0.552 | 0.588 | 0.564 | 0.532 | 0.552 | 0.588 | **0.707** | 0.656±0.009 |
| CricketX | 0.605 | 0.595 | 0.621 | 0.626 | 0.085 | 0.636 | 0.264 | 0.855 | **0.914**±0.009 |
| CricketY | 0.639 | 0.590 | 0.605 | 0.597 | 0.159 | 0.592 | 0.362 | **0.867** | 0.729±0.020 |
| CricketZ | 0.603 | 0.592 | 0.621 | 0.592 | 0.085 | 0.579 | 0.413 | **0.863** | 0.764±0.003 |
| DistalPhalanxOutlineC. | 0.775 | 0.775 | 0.754 | 0.739 | 0.699 | 0.696 | 0.750 | 0.766 | **0.804**±0.030 |
| DistalPhalanxTW | 0.662 | 0.698 | 0.669 | 0.674 | 0.676 | 0.647 | 0.691 | 0.664 | **0.781**±0.008 |
| Earthquakes | 0.748 | 0.748 | 0.748 | 0.748 | 0.655 | 0.770 | 0.670 | 0.670 | **0.784**±0.031 |
| ECG200 | 0.930 | 0.890 | 0.890 | 0.860 | 0.860 | 0.720 | 0.760 | 0.908 | **1.000**±0.000 |
| ECG5000 | 0.938 | 0.940 | 0.937 | 0.940 | 0.884 | 0.932 | 0.933 | 0.940 | **0.963**±0.012 |
| Faceall | 0.741 | 0.833 | 0.833 | 0.824 | 0.537 | 0.705 | 0.707 | 0.845 | **0.853**±0.007 |
| FacesUCR | 0.817 | 0.802 | 0.803 | 0.830 | 0.532 | 0.753 | 0.795 | **0.967** | 0.957±0.004 |
| FiftyWords | 0.686 | 0.686 | 0.697 | 0.688 | 0.343 | 0.602 | 0.653 | **0.816** | 0.699±0.024 |
| Fish | 0.903 | 0.840 | 0.857 | 0.880 | 0.280 | 0.383 | 0.240 | **0.987** | 0.920±0.037 |
| FreezerRegularTrain | 0.974 | 0.944 | 0.965 | 0.968 | 0.761 | 0.075 | 0.866 | 0.997 | **0.997**±0.036 |
| GunPoint | 0.980 | 0.953 | 0.953 | 0.940 | 0.820 | 0.747 | 0.807 | 0.999 | **1.000**±0.000 |
| GunPointAgeSpan | 0.965 | 0.946 | 0.959 | 0.940 | 0.478 | 0.478 | 0.956 | 0.992 | **1.000**±0.000 |
| GunPointMaleVSFemale | 0.991 | 0.997 | 0.994 | 0.997 | 0.687 | 0.525 | 0.997 | 0.999 | **1.000**±0.000 |
| GunPointOldVSYoung | 0.987 | 0.975 | 0.987 | 0.946 | 0.540 | 0.524 | 0.984 | **1.000** | **1.000**±0.000 |
| Ham | 0.705 | 0.716 | 0.667 | 0.714 | 0.533 | 0.600 | 0.610 | 0.704 | **0.733**±0.009 |
| Herring | 0.567 | 0.594 | 0.594 | 0.594 | 0.233 | 0.594 | 0.594 | 0.608 | **0.750**±0.014 |
| InsectEPGRegular | 0.996 | 0.992 | 0.996 | 0.968 | **1.000** | **1.000** | 0.984 | 0.951 | **1.000**±0.000 |
| Lightning2 | 0.787 | 0.738 | 0.705 | 0.689 | 0.459 | 0.705 | 0.672 | 0.807 | **0.819**±0.008 |
| Lightning7 | 0.616 | 0.603 | 0.630 | 0.603 | 0.233 | 0.699 | 0.767 | 0.793 | **0.808**±0.023 |
| Meat | 0.933 | 0.933 | 0.933 | 0.933 | 0.006 | 0.550 | 0.333 | 0.947 | **0.950**±0.016 |
| MedicalImages | 0.745 | 0.733 | 0.753 | 0.746 | 0.482 | 0.649 | 0.691 | **0.769** | 0.709±0.028 |
| MiddlePhalanxOutlineC. | 0.571 | 0.571 | 0.487 | 0.643 | 0.763 | 0.570 | 0.746 | 0.814 | **0.856**±0.001 |
| MiddlePhalanxTW | 0.578 | 0.610 | 0.597 | 0.604 | 0.584 | 0.584 | 0.591 | 0.519 | **0.642**±0.002 |
| OliveOil | **0.900** | 0.867 | 0.867 | 0.833 | 0.667 | 0.400 | 0.400 | 0.787 | 0.833±0.000 |
| plane | 0.981 | 0.962 | 0.971 | 0.971 | 0.962 | 0.848 | 0.962 | **1.000** | **1.000**±0.000 |
| PowerCons | 0.972 | 0.972 | 0.967 | 0.917 | 0.961 | 0.950 | **0.994** | 0.990 | 0.911±0.004 |
| ProximalPhalanxOutlineC. | 0.890 | 0.880 | 0.873 | 0.869 | 0.828 | 0.746 | 0.869 | 0.908 | **0.913**±0.006 |
| RefrigerationDevices | 0.469 | 0.371 | 0.365 | 0.411 | 0.360 | 0.509 | 0.467 | 0.503 | **0.613**±0.002 |
| ScreenType | 0.416 | 0.432 | 0.435 | 0.384 | 0.400 | 0.411 | 0.363 | **0.526** | 0.493±0.010 |
| SemgHandSubjectCh2 | 0.842 | 0.853 | 0.861 | 0.867 | 0.200 | 0.367 | 0.891 | 0.718 | **0.981**±0.025 |
| SmallKitchenAppliances | 0.675 | 0.384 | 0.403 | 0.379 | 0.602 | 0.760 | 0.715 | 0.721 | **0.803**±0.005 |
| SmoothSubspace | 0.960 | 0.873 | 0.920 | 0.867 | 0.940 | 0.953 | 0.927 | 0.989 | **1.000**±0.000 |
| StarLightCurves | 0.959 | 0.962 | 0.946 | 0.944 | 0.821 | 0.868 | 0.962 | 0.975 | **0.987**±0.014 |
| Strawberry | **0.984** | 0.976 | 0.970 | 0.968 | 0.943 | 0.754 | 0.916 | 0.982 | 0.919±0.000 |
| SwedishLeaf | 0.906 | 0.910 | 0.914 | 0.907 | 0.592 | 0.459 | 0.910 | **0.971** | 0.961±0.015 |
| SyntheticControl | 0.987 | 0.967 | 0.980 | 0.977 | 0.927 | 0.977 | 0.990 | 0.999 | **1.000**±0.000 |
| Trace | 0.960 | 0.810 | 0.760 | 0.760 | 0.700 | 0.710 | **1.000** | **1.000** | **1.000**±0.000 |
| TwoPatterns | 0.943 | 0.905 | 0.913 | 0.939 | 0.997 | 0.999 | **1.000** | **1.000** | **1.000**±0.000 |
| UMD | 0.917 | 0.924 | 0.972 | 0.910 | 0.444 | 0.715 | **1.000** | 0.993 | **1.000**±0.000 |
| UWaveGestureLibraryX | 0.796 | 0.787 | 0.785 | 0.658 | 0.560 | 0.753 | 0.736 | 0.822 | **0.832**±0.006 |
| UWaveGestureLibraryY | 0.716 | 0.706 | 0.704 | 0.703 | 0.445 | 0.652 | 0.654 | **0.757** | 0.754±0.003 |
| UWaveGestureLibraryZ | 0.740 | 0.739 | 0.729 | 0.719 | 0.433 | 0.678 | 0.703 | 0.764 | **0.805**±0.003 |
| WordSynonyms | 0.580 | 0.585 | 0.611 | 0.621 | 0.177 | 0.458 | 0.538 | 0.742 | **0.784**±0.022 |
| Worms | 0.571 | 0.507 | 0.558 | 0.507 | 0.351 | 0.494 | 0.416 | 0.765 | **0.803**±0.021 |
| WormsTwoClass | 0.623 | 0.623 | 0.610 | 0.597 | 0.519 | 0.468 | 0.571 | 0.657 | **0.815**±0.028 |
| Yoga | 0.849 | 0.846 | 0.846 | 0.849 | 0.464 | 0.767 | 0.618 | **0.911** | 0.772±0.006 |
| Average Accuracy | 80.1 | 77.7 | 78.2 | 77.7 | 56.0 | 63.1 | 69.5 | 84.8 | **87.0** |
| Average Rank | 4.2 | 5.0 | 4.9 | 5.4 | 7.8 | 7.2 | 6.1 | 2.7 | **1.8** |

Table G.3: ShapeConv Compared with State-Of-The-Art Time-Series Classification Methods: Evaluating Testing Accuracy for Supervised Time-Series Classification Tasks Across 128 UCR Datasets. Mean accuracy ± std over 3 independent experiments with different random seeds is reported.

| | DTW | TNC | TST | TS-TCC | T-Loss | TS2Vec | ROCKET | ShapeConv |
|---|---|---|---|---|---|---|---|---|
| Adiac | 0.604 | 0.726 | 0.550 | 0.767 | 0.675 | 0.775 | 0.783 | **0.867**±0.000 |
| ArrowHead | 0.703 | 0.703 | 0.771 | 0.737 | 0.766 | 0.857 | 0.814 | **0.903**±0.005 |
| Beef | 0.633 | 0.733 | 0.500 | 0.600 | 0.667 | 0.767 | 0.833 | **0.936**±0.014 |
| BeetleFly | 0.700 | 0.850 | **1.000** | 0.800 | 0.800 | 0.900 | 0.900 | **1.000**±0.000 |
| BirdChicken | 0.750 | 0.750 | 0.650 | 0.650 | 0.850 | 0.800 | 0.900 | **1.000**±0.000 |
| Car | 0.733 | 0.683 | 0.550 | 0.583 | 0.833 | 0.883 | 0.847 | **0.974**±0.000 |
| CBF | 0.997 | 0.983 | 0.898 | 0.998 | 0.983 | **1.000** | **1.000** | **1.000**±0.000 |
| ChlorineConcentration | 0.648 | 0.760 | 0.562 | 0.753 | 0.749 | 0.832 | 0.815 | **0.924**±0.003 |
| CinCECGTorso | 0.651 | 0.669 | 0.508 | 0.671 | 0.713 | 0.827 | **0.836** | 0.778±0.007 |
| Coffee | **1.000** | **1.000** | 0.821 | **1.000** | **1.000** | **1.000** | **1.000** | **1.000**±0.000 |
| Computers | 0.700 | 0.684 | 0.696 | 0.704 | 0.664 | 0.660 | **0.761** | 0.647±0.000 |
| CricketX | 0.754 | 0.623 | 0.385 | 0.731 | 0.713 | 0.805 | 0.819 | **0.895**±0.009 |
| CricketY | 0.744 | 0.597 | 0.467 | 0.718 | 0.728 | 0.769 | **0.852** | 0.726±0.016 |
| CricketZ | 0.754 | 0.682 | 0.403 | 0.713 | 0.708 | 0.792 | **0.856** | 0.773±0.017 |
| DiatomSizeReduction | 0.967 | 0.993 | 0.961 | 0.977 | 0.984 | 0.987 | 0.970 | **0.994**±0.005 |
| DistalPhalanxOutlineCorrect | 0.717 | 0.754 | 0.728 | 0.754 | **0.775** | **0.775** | 0.770 | 0.753±0.026 |
| DistalPhalanxOutlineAgeGroup | 0.770 | 0.741 | 0.741 | 0.755 | 0.727 | 0.727 | 0.759 | **0.784**±0.017 |
| DistalPhalanxTW | 0.590 | 0.669 | 0.568 | 0.676 | 0.676 | 0.698 | 0.719 | **0.763**±0.022 |
| Earthquakes | 0.719 | 0.748 | 0.748 | 0.748 | 0.748 | 0.748 | **0.748** | 0.731±0.010 |
| ECG200 | 0.770 | 0.830 | 0.830 | 0.880 | 0.940 | 0.920 | 0.906 | **0.992**±0.001 |
| ECG5000 | 0.924 | 0.937 | 0.928 | 0.941 | 0.933 | 0.935 | 0.947 | **0.953**±0.007 |
| ECGFiveDays | 0.768 | 0.999 | 0.763 | 0.878 | **1.000** | **1.000** | **1.000** | **1.000**±0.000 |
| ElectricDevices | 0.602 | 0.700 | 0.676 | 0.686 | 0.707 | 0.721 | 0.729 | **0.743**±0.013 |
| FaceAll | 0.808 | 0.766 | 0.504 | 0.813 | 0.786 | 0.805 | **0.947** | 0.827±0.037 |
| FaceFour | 0.830 | 0.659 | 0.511 | 0.773 | 0.920 | 0.932 | **0.977** | 0.961±0.019 |
| FacesUCR | 0.905 | 0.789 | 0.543 | 0.863 | 0.884 | 0.930 | **0.961** | 0.930±0.000 |
| FiftyWords | 0.690 | 0.653 | 0.525 | 0.653 | 0.732 | 0.774 | **0.830** | 0.699±0.009 |
| Fish | 0.823 | 0.817 | 0.720 | 0.817 | 0.891 | 0.937 | **0.979** | 0.917±0.006 |
| FordA | 0.555 | 0.902 | 0.568 | 0.930 | 0.928 | 0.948 | 0.944 | **0.954**±0.020 |
| FordB | 0.620 | 0.733 | 0.507 | 0.815 | 0.793 | 0.807 | 0.805 | **0.835**±0.022 |
| GunPoint | 0.907 | 0.967 | 0.827 | 0.993 | 0.980 | 0.987 | **1.000** | 0.997±0.002 |
| Ham | 0.467 | **0.752** | 0.524 | 0.743 | 0.724 | 0.724 | 0.726 | 0.733±0.032 |
| HandOutlines | 0.881 | 0.930 | 0.735 | 0.724 | 0.922 | 0.930 | 0.942 | **0.947**±0.016 |
| Haptics | 0.377 | 0.474 | 0.357 | 0.396 | 0.490 | 0.536 | 0.524 | **0.580**±0.003 |
| Herring | 0.531 | 0.594 | 0.594 | 0.594 | 0.594 | 0.641 | 0.692 | **0.724**±0.008 |
| InlineSkate | 0.384 | 0.378 | 0.287 | 0.347 | 0.371 | 0.415 | **0.457** | 0.432±0.023 |
| InsectWingbeatSound | 0.355 | 0.549 | 0.266 | 0.415 | 0.597 | 0.630 | **0.657** | 0.613±0.008 |
| ItalyPowerDemand | 0.950 | 0.928 | 0.845 | 0.955 | 0.954 | 0.961 | 0.970 | **0.974**±0.017 |
| LargeKitchenAppliances | 0.795 | 0.776 | 0.595 | 0.848 | 0.789 | 0.875 | 0.901 | **0.917**±0.003 |
| Lightning2 | **0.869** | **0.869** | 0.705 | 0.836 | **0.869** | **0.869** | 0.759 | 0.819±0.000 |
| Lightning7 | 0.726 | 0.767 | 0.411 | 0.685 | 0.795 | **0.863** | 0.823 | 0.781±0.007 |
| Mallat | 0.934 | 0.871 | 0.713 | 0.922 | 0.951 | 0.915 | **0.956** | 0.932±0.010 |
| Meat | 0.933 | 0.917 | 0.900 | 0.883 | 0.950 | **0.967** | 0.948 | 0.943±0.017 |
| MedicalImages | 0.737 | 0.754 | 0.632 | 0.747 | 0.750 | 0.793 | **0.799** | 0.774±0.003 |
| MiddlePhalanxOutlineCorrect | 0.698 | 0.818 | 0.753 | 0.818 | 0.825 | 0.838 | **0.838** | 0.827±0.004 |
| MiddlePhalanxOutlineAgeGroup | 0.500 | 0.643 | 0.617 | 0.630 | 0.656 | 0.636 | 0.590 | **0.669**±0.011 |
| MiddlePhalanxTW | 0.506 | 0.571 | 0.506 | 0.610 | 0.591 | 0.591 | 0.560 | **0.637**±0.007 |
| MoteStrain | 0.835 | 0.825 | 0.768 | 0.843 | 0.851 | 0.863 | 0.915 | **0.919**±0.029 |
| NonInvasiveFetalECGThorax1 | 0.790 | 0.898 | 0.471 | 0.898 | 0.878 | **0.930** | 0.913 | 0.913±0.012 |
| NonInvasiveFetalECGThorax2 | 0.865 | 0.912 | 0.832 | 0.913 | 0.919 | 0.940 | 0.929 | **0.942**±0.013 |
| OliveOil | 0.833 | 0.833 | 0.800 | 0.800 | 0.867 | 0.900 | **0.917** | 0.827±0.015 |
| OSULeaf | 0.591 | 0.723 | 0.545 | 0.723 | 0.760 | 0.876 | **0.941** | 0.905±0.004 |
| PhalangesOutlinesCorrect | 0.728 | 0.787 | 0.773 | 0.804 | 0.784 | 0.823 | **0.834** | 0.813±0.017 |
| Phoneme | 0.228 | 0.180 | 0.139 | 0.242 | 0.276 | **0.312** | 0.280 | 0.204±0.004 |
| Plane | **1.000** | **1.000** | 0.933 | **1.000** | 0.990 | **1.000** | **1.000** | **1.000**±0.000 |
| ProximalPhalanxOutlineCorrect | 0.784 | 0.866 | 0.770 | 0.873 | 0.859 | 0.900 | 0.899 | **0.913**±0.019 |
| ProximalPhalanxOutlineAgeGroup | 0.805 | 0.854 | 0.854 | 0.839 | 0.844 | 0.844 | 0.856 | **0.869**±0.004 |
| ProximalPhalanxTW | 0.761 | 0.810 | 0.780 | 0.800 | 0.771 | 0.824 | 0.817 | **0.831**±0.033 |
| RefrigerationDevices | 0.464 | 0.565 | 0.483 | 0.563 | 0.515 | 0.589 | 0.537 | **0.594**±0.026 |
| ScreenType | 0.397 | **0.509** | 0.419 | 0.419 | 0.416 | 0.411 | 0.485 | 0.423±0.007 |
| ShapeletSim | 0.650 | 0.589 | 0.489 | 0.683 | 0.672 | **1.000** | **1.000** | **1.000**±0.000 |
| ShapesAll | 0.768 | 0.788 | 0.733 | 0.773 | 0.848 | 0.905 | **0.907** | 0.853±0.001 |
| SmallKitchenAppliances | 0.643 | 0.725 | 0.592 | 0.691 | 0.677 | 0.733 | **0.818** | 0.741±0.005 |
| SonyAIBORobotSurface1 | 0.725 | 0.804 | 0.724 | 0.899 | 0.902 | 0.903 | 0.922 | **0.962**±0.005 |
| SonyAIBORobotSurface2 | 0.831 | 0.834 | 0.745 | 0.907 | 0.889 | 0.890 | 0.913 | **0.914**±0.018 |
| StarLightCurves | 0.907 | 0.968 | 0.949 | 0.967 | 0.964 | 0.971 | 0.981 | **0.987**±0.004 |
| Strawberry | 0.941 | 0.951 | 0.916 | 0.965 | 0.954 | 0.965 | **0.981** | 0.903±0.008 |
| SwedishLeaf | 0.792 | 0.880 | 0.738 | 0.923 | 0.914 | 0.942 | **0.964** | 0.952±0.012 |
| Symbols | 0.950 | 0.885 | 0.786 | 0.916 | 0.963 | 0.976 | 0.974 | **0.980**±0.041 |
| SyntheticControl | 0.993 | **1.000** | 0.490 | 0.990 | 0.987 | 0.997 | 1.000 | **1.000**±0.000 |
| ToeSegmentation1 | 0.772 | 0.864 | 0.807 | 0.930 | 0.939 | 0.947 | **0.968** | 0.957±0.003 |
| ToeSegmentation2 | 0.838 | 0.831 | 0.615 | 0.877 | 0.900 | 0.915 | 0.924 | **0.931**±0.020 |
| Trace | **1.000** | **1.000** | **1.000** | **1.000** | 0.990 | **1.000** | **1.000** | **1.000**±0.000 |

| | | | | | | | | |
|---|---|---|---|---|---|---|---|---|
| TwoLeadECG | 0.905 | 0.993 | 0.871 | 0.976 | 0.999 | 0.987 | 0.999 | **1.000**±0.000 |
| TwoPatterns | **1.000** | **1.000** | 0.466 | 0.999 | 0.999 | **1.000** | **1.000** | **1.000**±0.000 |
| UWaveGestureLibraryX | 0.728 | 0.781 | 0.569 | 0.733 | 0.785 | 0.810 | **0.815** | 0.805±0.034 |
| UWaveGestureLibraryY | 0.634 | 0.697 | 0.348 | 0.641 | 0.710 | 0.729 | **0.744** | 0.738±0.013 |
| UWaveGestureLibraryZ | 0.658 | 0.721 | 0.655 | 0.690 | 0.757 | 0.770 | 0.732 | **0.792**±0.018 |
| UWaveGestureLibraryAll | 0.892 | 0.903 | 0.475 | 0.692 | 0.896 | 0.934 | 0.925 | **0.941**±0.028 |
| Wafer | 0.980 | 0.994 | 0.991 | 0.994 | 0.992 | 0.998 | **0.998** | 0.973±0.006 |
| Wine | 0.574 | 0.759 | 0.500 | 0.778 | 0.815 | 0.889 | 0.813 | **0.894**±0.018 |
| WordSynonyms | 0.649 | 0.630 | 0.422 | 0.531 | 0.691 | 0.704 | 0.753 | **0.765**±0.025 |
| Worms | 0.584 | 0.623 | 0.455 | 0.753 | 0.727 | 0.701 | 0.740 | **0.783**±0.002 |
| WormsTwoClass | 0.623 | 0.727 | 0.584 | 0.753 | 0.792 | 0.805 | 0.797 | **0.815**±0.011 |
| Yoga | 0.837 | 0.812 | 0.830 | 0.791 | 0.837 | 0.887 | **0.910** | 0.742±0.047 |
| ACSF1 | 0.640 | 0.730 | 0.760 | 0.730 | 0.900 | **0.910** | 0.886 | 0.902±0.020 |
| AllGestureWiimoteX | 0.716 | 0.703 | 0.259 | 0.697 | 0.763 | 0.777 | 0.790 | **0.831**±0.042 |
| AllGestureWiimoteY | 0.729 | 0.699 | 0.423 | 0.741 | 0.726 | 0.793 | 0.773 | **0.826**±0.017 |
| AllGestureWiimoteZ | 0.643 | 0.646 | 0.447 | 0.689 | 0.723 | 0.770 | 0.766 | **0.848**±0.002 |
| BME | 0.900 | 0.973 | 0.760 | 0.933 | 0.993 | 0.993 | **1.000** | **1.000**±0.000 |
| Chinatown | 0.957 | 0.977 | 0.936 | **0.983** | 0.951 | 0.968 | 0.983 | 0.954±0.016 |
| Crop | 0.665 | 0.738 | 0.710 | 0.742 | 0.722 | **0.756** | 0.751 | 0.703±0.005 |
| EOGHorizontalSignal | 0.503 | 0.442 | 0.373 | 0.401 | 0.605 | 0.544 | 0.539 | **0.609**±0.018 |
| EOGVerticalSignal | 0.448 | 0.392 | 0.298 | 0.376 | 0.434 | 0.503 | 0.441 | **0.521**±0.013 |
| EthanolLevel | 0.276 | 0.424 | 0.260 | 0.486 | 0.382 | 0.484 | 0.583 | **0.704**±0.018 |
| FreezerRegularTrain | 0.899 | 0.991 | 0.922 | 0.989 | 0.956 | 0.986 | **0.998** | 0.993±0.003 |
| FreezerSmallTrain | 0.753 | **0.982** | 0.920 | 0.979 | 0.933 | 0.894 | 0.950 | 0.972±0.012 |
| Fungi | 0.839 | 0.527 | 0.366 | 0.753 | **1.000** | 0.962 | **1.000** | 0.954±0.003 |
| GestureMidAirD1 | 0.569 | 0.431 | 0.208 | 0.369 | 0.608 | **0.631** | 0.617 | 0.541±0.026 |
| GestureMidAirD2 | **0.608** | 0.362 | 0.138 | 0.254 | 0.546 | 0.515 | 0.561 | 0.585±0.029 |
| GestureMidAirD3 | 0.323 | 0.292 | 0.154 | 0.177 | 0.285 | 0.346 | 0.315 | **0.405**±0.008 |
| GesturePebbleZ1 | 0.791 | 0.378 | 0.500 | 0.395 | 0.919 | **0.930** | 0.906 | 0.871±0.002 |
| GesturePebbleZ2 | 0.671 | 0.316 | 0.380 | 0.430 | **0.899** | 0.873 | 0.830 | 0.874±0.030 |
| GunPointAgeSpan | 0.918 | 0.984 | 0.991 | 0.994 | 0.994 | 0.994 | 0.997 | **1.000**±0.000 |
| GunPointMaleVersusFemale | 0.997 | 0.994 | **1.000** | 0.997 | 0.997 | **1.000** | 0.998 | **1.000**±0.000 |
| GunPointOldVersusYoung | 0.838 | **1.000** | **1.000** | **1.000** | **1.000** | **1.000** | 0.991 | **1.000**±0.000 |
| HouseTwenty | 0.924 | 0.782 | 0.815 | 0.790 | 0.933 | 0.941 | **0.964** | 0.953±0.018 |
| InsectEPGRegularTrain | 0.872 | **1.000** | **1.000** | **1.000** | **1.000** | **1.000** | **1.000** | **1.000**±0.000 |
| InsectEPGSmallTrain | 0.735 | **1.000** | **1.000** | **1.000** | **1.000** | **1.000** | 0.979 | **1.000**±0.000 |
| MelbournePedestrian | 0.791 | 0.942 | 0.741 | 0.949 | 0.944 | **0.959** | 0.904 | 0.926±0.016 |
| MixedShapesRegularTrain | 0.842 | 0.911 | 0.879 | 0.855 | 0.905 | 0.922 | 0.921 | **0.965**±0.021 |
| MixedShapesSmallTrain | 0.780 | 0.813 | 0.828 | 0.735 | 0.860 | 0.881 | 0.918 | **0.927**±0.011 |
| PickupGestureWiimoteZ | 0.660 | 0.620 | 0.240 | 0.600 | 0.740 | 0.820 | 0.830 | **0.871**±0.026 |
| PigAirwayPressure | 0.106 | 0.413 | 0.120 | 0.380 | 0.510 | **0.683** | 0.095 | 0.594±0.005 |
| PigArtPressure | 0.245 | 0.808 | 0.774 | 0.524 | 0.928 | **0.966** | 0.954 | 0.872±0.022 |
| PigCVP | 0.154 | 0.649 | 0.596 | 0.615 | 0.788 | 0.870 | **0.934** | 0.831±0.006 |
| PLAID | 0.840 | 0.495 | 0.419 | 0.445 | 0.555 | 0.561 | 0.903 | **0.904**±0.008 |
| PowerCons | 0.878 | 0.933 | 0.911 | 0.961 | 0.900 | **0.972** | 0.940 | 0.901±0.000 |
| Rock | 0.600 | 0.580 | 0.680 | 0.600 | 0.580 | 0.700 | **0.900** | 0.700±0.016 |
| SemgHandGenderCh2 | 0.802 | 0.882 | 0.725 | 0.837 | 0.890 | 0.963 | 0.927 | **0.972**±0.004 |
| SemgHandMovementCh2 | 0.584 | 0.593 | 0.420 | 0.613 | 0.789 | 0.893 | 0.645 | **0.924**±0.005 |
| SemgHandSubjectCh2 | 0.727 | 0.771 | 0.484 | 0.753 | 0.853 | 0.951 | 0.881 | **0.981**±0.002 |
| ShakeGestureWiimoteZ | 0.860 | 0.820 | 0.760 | 0.860 | 0.920 | **0.940** | 0.898 | 0.834±0.001 |
| SmoothSubspace | 0.827 | 0.913 | 0.827 | 0.953 | 0.960 | 0.993 | 0.979 | **1.000**±0.000 |
| UMD | 0.993 | 0.993 | 0.910 | 0.986 | 0.993 | **1.000** | 0.992 | **1.000**±0.000 |
| DodgerLoopDay | 0.500 | 0.000 | 0.200 | 0.000 | 0.000 | 0.562 | 0.573 | **0.628**±0.003 |
| DodgerLoopGame | 0.877 | 0.000 | 0.696 | 0.000 | 0.000 | 0.841 | 0.873 | **0.906**±0.006 |
| DodgerLoopWeekend | 0.949 | 0.000 | 0.732 | 0.000 | 0.000 | 0.964 | **0.975** | 0.971±0.022 |
| Average Accuarcy | 0.728 | 0.743 | 0.639 | 0.740 | 0.787 | 0.836 | 0.842 | **0.851** |
| Average Rank | 6.102 | 5.285 | 7.055 | 5.191 | 4.523 | 2.855 | 2.590 | **2.398** |

Table G.4: ShapeConv Compared with State-Of-The-Art Time-Series Classification Methods: Evaluating Testing Accuracy for Supervised Multivariate Time-Series Classification Tasks Across 30 UEA Datasets. Mean accuracy ± std over 3 independent experiments with different random seeds is reported.

| | DTW | TNC | TST | TS-TCC | T-Loss | TS2Vec | ShapeConv |
|---|---|---|---|---|---|---|---|
| ArticularyWordRecognition | 0.987 | 0.973 | 0.977 | 0.953 | 0.943 | 0.987 | **0.994**±0.001 |
| AtrialFibrillation | 0.200 | 0.133 | 0.067 | 0.267 | 0.133 | 0.200 | **0.521**±0.015 |
| BasicMotions | 0.975 | 0.975 | 0.975 | **1.000** | **1.000** | 0.975 | 0.997±0.016 |
| CharacterTrajectories | 0.989 | 0.967 | 0.975 | 0.985 | 0.993 | **0.995** | 0.981±0.018 |
| Cricket | **1.000** | 0.958 | **1.000** | 0.917 | 0.972 | 0.972 | 0.998±0.008 |
| DuckDuckGeese | 0.600 | 0.460 | 0.620 | 0.380 | 0.650 | **0.680** | 0.648±0.006 |
| EigenWorms | 0.618 | 0.840 | 0.748 | 0.779 | 0.840 | **0.847** | 0.802±0.008 |
| Epilepsy | 0.964 | 0.957 | 0.949 | 0.957 | 0.971 | 0.964 | **0.972**±0.009 |
| ERing | 0.133 | 0.852 | 0.874 | **0.904** | 0.133 | 0.874 | 0.774±0.003 |
| EthanolConcentration | **0.323** | 0.297 | 0.262 | 0.285 | 0.205 | 0.308 | 0.253±0.001 |
| FaceDetection | 0.529 | 0.536 | 0.534 | 0.544 | 0.513 | 0.501 | **0.635**±0.025 |
| FingerMovements | 0.530 | 0.470 | 0.560 | 0.460 | 0.580 | 0.480 | **0.587**±0.029 |
| HandMovementDirection | 0.231 | 0.324 | 0.243 | 0.243 | 0.351 | 0.338 | **0.413**±0.020 |
| Handwriting | 0.286 | 0.249 | 0.225 | 0.498 | 0.451 | 0.515 | **0.527**±0.006 |
| Heartbeat | 0.717 | 0.746 | 0.746 | 0.751 | 0.741 | 0.683 | **0.784**±0.011 |
| JapaneseVowels | 0.949 | 0.978 | 0.978 | 0.930 | 0.989 | 0.984 | **0.993**±0.022 |
| Libras | 0.870 | 0.817 | 0.656 | 0.822 | 0.883 | 0.867 | **0.887**±0.002 |
| LSST | 0.551 | 0.595 | 0.408 | 0.474 | 0.509 | 0.537 | **0.608**±0.023 |
| MotorImagery | 0.500 | 0.500 | 0.500 | 0.610 | 0.580 | 0.510 | **0.674**±0.002 |
| NATOPS | 0.883 | 0.911 | 0.850 | 0.822 | 0.917 | 0.928 | **0.937**±0.004 |
| PEMS-SF | 0.711 | 0.699 | 0.740 | 0.734 | 0.676 | 0.682 | **0.801**±0.005 |
| PenDigits | 0.977 | 0.979 | 0.560 | 0.974 | 0.981 | **0.989** | 0.968±0.018 |
| PhonemeSpectra | 0.151 | 0.207 | 0.085 | **0.252** | 0.222 | 0.233 | 0.192±0.002 |
| RacketSports | 0.803 | 0.776 | 0.809 | 0.816 | 0.855 | 0.855 | **0.863**±0.012 |
| SelfRegulationSCP1 | 0.775 | 0.799 | 0.754 | 0.823 | 0.843 | 0.812 | **0.858**±0.003 |
| SelfRegulationSCP2 | 0.539 | 0.550 | 0.550 | 0.533 | 0.539 | 0.578 | **0.624**±0.054 |
| SpokenArabicDigits | 0.963 | 0.934 | 0.923 | 0.970 | 0.905 | **0.988** | 0.979±0.020 |
| StandWalkJump | 0.200 | 0.400 | 0.267 | 0.333 | 0.333 | 0.467 | **0.587**±0.013 |
| UWaveGestureLibrary | 0.903 | 0.759 | 0.575 | 0.753 | 0.875 | 0.906 | **0.936**±0.002 |
| InsectWingbeat | - | 0.469 | 0.105 | 0.264 | 0.156 | 0.466 | **0.509**±0.026 |
| Average Accuracy | 0.650 | 0.670 | 0.617 | 0.668 | 0.658 | 0.704 | **0.743** |
| Average Rank | 4.717 | 4.583 | 5.283 | 4.333 | 3.900 | 3.117 | **2.067** |

## G.4 RESULTS OF UNSUPERVISED LEARNING TASKS

In this section, we report the full results of unsupervised time-series clustering tasks across 36 UCR Datasets in Table G.5. We also report the result of unsupervised multivariate time-series clustering tasks compared with Zhang & Sun (2022) across 12 UEA datasets in Table G.6.

Table G.5: Unsupervised time-series clustering results (NMI on test data) across 36 UCR datasets. Mean accuracy ± std over 3 independent experiments with different random seeds is reported.

| Dataset | KMeans | k-Shape | U-ShapeL | DTC | USSL | DTCR | STCN | AutoS. | ShapeC. w/o Init | ShapeC. w/o DBI | ShapeC. | ShapeC. w/ Human |
|---|---|---|---|---|---|---|---|---|---|---|---|---|
| Arrow | 0.4816 | 0.5240 | 0.3522 | 0.5000 | **0.6322** | 0.5513 | 0.5240 | 0.5624 | 0.5134 | 0.6123 | 0.6064±0.010 | **0.6445**±0.013 |
| Beef | 0.2925 | 0.3338 | 0.3413 | 0.2751 | 0.3338 | **0.5473** | 0.5432 | 0.3799 | 0.3077 | 0.3854 | 0.4039±0.003 | **0.6188**±0.017 |
| BeetleFly | 0.0073 | 0.3456 | 0.5105 | 0.3456 | 0.5310 | 0.7610 | **1.0000** | 0.5310 | 0.4897 | **1.0000** | **1.0000**±0.000 | **1.0000**±0.000 |
| BirdChicken | 0.0371 | 0.3456 | 0.2783 | 0.0073 | 0.6190 | 0.5310 | **1.0000** | 0.6352 | 0.5824 | **1.0000** | **1.0000**±0.000 | **1.0000**±0.000 |
| Car | 0.2540 | 0.3771 | 0.3655 | 0.1892 | 0.4650 | 0.5021 | **0.5701** | 0.4970 | 0.3672 | 0.4690 | 0.4770±0.006 | 0.5013±0.013 |
| Chlorine. | 0.0129 | 0.0000 | 0.0135 | 0.0013 | 0.0133 | 0.0195 | **0.0760** | 0.0133 | 0.0024 | 0.0368 | 0.0527±0.007 | 0.0641±0.014 |
| Coffee | 0.5246 | **1.0000** | **1.0000** | 0.5523 | **1.0000** | 0.6277 | **1.0000** | **1.0000** | **1.0000** | **1.0000** | **1.0000**±0.000 | **1.0000**±0.000 |
| Diatom. | 0.9300 | **1.0000** | 0.4849 | 0.6863 | **1.0000** | 0.9418 | **1.0000** | **1.0000** | **1.0000** | **1.0000** | **1.0000**±0.000 | **1.0000**±0.000 |
| Dist.ageG | 0.1880 | 0.2911 | 0.2577 | 0.3406 | 0.3846 | 0.4553 | **0.5037** | 0.4400 | 0.4237 | 0.4464 | 0.4786±0.004 | **0.5291**±0.007 |
| Dist.correct | 0.0278 | 0.0527 | 0.0063 | 0.0115 | 0.1026 | 0.1180 | **0.2327** | 0.1333 | 0.0699 | 0.0885 | 0.1074±0.009 | 0.1836±0.012 |
| ECG200 | 0.1403 | 0.3682 | 0.1323 | 0.0918 | 0.3776 | 0.3691 | 0.4316 | 0.3928 | 0.3002 | 0.5413 | **0.5552**±0.009 | **0.6240**±0.014 |
| ECGFiveDays | 0.0002 | 0.0002 | 0.1498 | 0.0022 | 0.6502 | 0.8056 | 0.7835 | 0.6355 | 0.8150 | **0.8246**±0.010 | 0.7669±0.027 |
| GunPoint | 0.0126 | 0.3653 | 0.3653 | 0.0194 | 0.4878 | 0.4200 | **0.5537** | 0.4027 | 0.3803 | 0.4248 | 0.4476±0.003 | **0.5652**±0.025 |
| Ham | 0.0093 | 0.0517 | 0.0619 | 0.1016 | 0.3411 | 0.0989 | 0.2382 | 0.3211 | 0.1764 | 0.3859 | **0.3911**±0.001 | **0.4467**±0.005 |
| Herring | 0.0013 | 0.0027 | 0.1324 | 0.0143 | 0.1718 | 0.2248 | 0.2002 | 0.2019 | 0.1423 | 0.2293 | **0.2630**±0.012 | 0.2317±0.014 |
| Lighting2 | 0.0038 | 0.2670 | 0.0144 | 0.1435 | 0.3727 | 0.2289 | 0.3479 | 0.3530 | 0.3040 | 0.3756 | **0.4282**±0.008 | **0.4723**±0.002 |
| Meat | 0.2510 | 0.2254 | 0.2716 | 0.2250 | 0.9085 | **0.9653** | 0.9393 | 0.9437 | 0.2901 | 0.9423 | 0.9481±0.000 | **1.0000**±0.000 |
| Mid.ageG | 0.0219 | 0.0722 | 0.1491 | 0.1390 | 0.2780 | 0.4661 | **0.5109** | 0.3940 | 0.1830 | 0.4214 | 0.4576±0.003 | **0.6654**±0.013 |
| Mid.correct | 0.0024 | 0.0349 | 0.0253 | 0.0079 | 0.2503 | 0.1150 | 0.0921 | **0.2873** | 0.1942 | 0.2109 | 0.2096±0.026 | **0.3499**±0.002 |
| Mid.TW | 0.4134 | 0.5229 | 0.4065 | 0.1156 | 0.9202 | 0.5503 | 0.6169 | **0.9450** | 0.7836 | 0.9161 | 0.9241±0.004 | 0.9310±0.006 |
| MoteStrain | 0.0551 | 0.2215 | 0.0082 | 0.0094 | 0.5310 | 0.4094 | 0.4063 | 0.4257 | 0.2976 | 0.5406 | **0.5738**±0.001 | 0.5982±0.014 |
| OSULeaf | 0.0208 | 0.0126 | 0.0203 | 0.2201 | 0.3353 | 0.2599 | 0.3544 | 0.4432 | 0.2952 | 0.5131 | **0.5159**±0.017 | 0.4846±0.018 |
| Plane | 0.8598 | 0.9642 | **1.0000** | 0.8678 | 0.9296 | 0.9615 | 0.9982 | **1.0000** | **1.0000**±0.000 | **1.0000**±0.000 |
| Prox.ageG | 0.0635 | 0.0110 | 0.0332 | 0.4153 | 0.6813 | 0.5581 | 0.6317 | **0.6930** | 0.5164 | 0.6057 | 0.6453±0.013 | **0.7292**±0.013 |
| Prox.TW | 0.0082 | 0.1577 | 0.0107 | 0.6199 | **1.0000** | 0.6539 | 0.7330 | 0.8947 | 0.5948 | 0.8368 | 0.8284±0.006 | 0.8117±0.005 |
| Sony. | 0.6112 | **0.7107** | 0.5803 | 0.2559 | 0.5597 | 0.6634 | 0.6112 | 0.6096 | 0.5423 | 0.6256 | 0.6215±0.047 | 0.6107±0.005 |
| Sony.II | 0.5444 | 0.0110 | 0.5903 | 0.4257 | 0.6858 | 0.6121 | 0.5647 | **0.7020** | 0.4985 | 0.6765 | 0.6767±0.015 | **0.7343**±0.034 |
| SwedishLeaf | 0.0168 | 0.1041 | 0.3456 | 0.6187 | 0.9186 | 0.6663 | 0.6106 | **0.9340** | 0.5834 | 0.8592 | 0.8418±0.006 | 0.8412±0.010 |
| Symbols | 0.7780 | 0.6366 | 0.8691 | 0.7995 | 0.8821 | 0.8989 | 0.8940 | 0.9147 | 0.7778 | **0.9313** | 0.9250±0.030 | 0.9118±0.002 |
| ToeSeg.1 | 0.0022 | 0.3073 | 0.3073 | 0.0188 | 0.3351 | 0.3115 | 0.3671 | 0.4610 | 0.2830 | 0.4700 | **0.4863**±0.001 | **0.5851**±0.007 |
| ToeSeg.2 | 0.0863 | 0.0863 | 0.1519 | 0.0096 | 0.4308 | 0.3249 | **0.5498** | 0.4664 | 0.1293 | 0.4959 | 0.5178±0.002 | **0.6636**±0.007 |
| TwoPatterns | 0.4696 | 0.3949 | 0.2979 | 0.0119 | 0.4911 | 0.4713 | 0.4110 | 0.5150 | 0.3083 | 0.5030 | **0.5177**±0.004 | **0.6154**±0.019 |
| TwoLeadECG | 0.0000 | 0.0000 | 0.0529 | 0.0036 | 0.5471 | 0.4614 | **0.6911** | 0.5654 | 0.4220 | 0.6006 | 0.6289±0.033 | **0.7045**±0.001 |
| Wafer | 0.0010 | 0.0010 | 0.0010 | 0.0008 | 0.0492 | 0.0228 | **0.2089** | 0.0520 | -0.0063 | 0.0741 | 0.0802±0.008 | 0.0477±0.021 |
| Wine | 0.0031 | 0.0119 | 0.0171 | 0.0000 | **0.7511** | 0.2580 | 0.5927 | 0.6045 | 0.2840 | 0.6090 | 0.6328±0.016 | 0.6710±0.003 |
| WordsS. | 0.5435 | 0.4154 | 0.3933 | 0.3498 | 0.4984 | 0.5448 | 0.3947 | 0.5112 | 0.3861 | 0.5884 | **0.5952**±0.003 | **0.6654**±0.001 |
| Avg. Acc. | 0.2132 | 0.2841 | 0.2777 | 0.2332 | 0.5427 | 0.4818 | 0.5478 | 0.5558 | 0.4183 | 0.5897 | **0.6017** | **0.6464** |
| Avg. Rank | 9.5556 | 8.4028 | 8.4306 | 9.6250 | 4.3611 | 5.0833 | 4.1250 | 3.7083 | 7.2639 | 3.0694 | **2.3750** | NA |

Table G.6: Unsupervised time-series clustering results (NMI on test data) across 12 UEA datasets. Mean accuracy ± std over 3 independent experiments with different random seeds is reported.

| Dataset | MC2PCA | SWIMDFC | TCK | m-kAVG+ED | m-kDBA | m-kShape | m-kSC | DeTSEC | NESE | MUSLA | ShapeConv |
|---|---|---|---|---|---|---|---|---|---|---|---|
| ArticularyWordR. | **0.934** | 0.523 | 0.873 | 0.834 | 0.741 | 0.344 | 0.843 | 0.792 | 0.849 | 0.838 | 0.867±0.010 |
| AtrialFibrilation | 0.514 | 0.532 | 0.191 | 0.515 | 0.317 | 0.116 | 0.387 | 0.293 | 0.346 | 0.538 | **0.579**±0.015 |
| BasicMotions | 0.674 | 0.510 | 0.776 | 0.543 | 0.639 | 0.341 | 0.554 | 0.800 | 0.525 | **1.000** | **1.000**±0.000 |
| Epilepsy | 0.173 | 0.190 | 0.533 | 0.409 | 0.471 | 0.163 | 0.381 | 0.345 | **0.760** | 0.601 | 0.681±0.003 |
| Ering | 0.336 | 0.422 | 0.399 | 0.400 | 0.406 | 0.268 | 0.348 | 0.392 | 0.378 | 0.722 | **0.736**±0.004 |
| HandMovementD. | 0.067 | 0.151 | 0.103 | 0.168 | 0.265 | 0.079 | 0.151 | 0.112 | 0.030 | **0.398** | 0.362±0.011 |
| Libras | 0.577 | 0.500 | 0.620 | 0.622 | 0.622 | 0.447 | 0.724 | 0.602 | 0.542 | 0.724 | **0.738**±0.004 |
| NATOPS | 0.698 | 0.472 | 0.679 | 0.643 | 0.643 | 0.339 | 0.600 | 0.043 | 0.314 | 0.855 | **0.878**±0.007 |
| PEMS-SF | 0.011 | 0.441 | 0.066 | 0.491 | 0.402 | 0.447 | 0.474 | 0.424 | 0.586 | 0.614 | **0.630**±0.004 |
| PenDigits | 0.713 | 0.652 | 0.693 | 0.738 | 0.605 | 0.634 | 0.738 | 0.563 | 0.645 | **0.826** | 0.784±0.004 |
| StandWalkJump | 0.349 | 0.483 | 0.536 | 0.559 | 0.466 | 0.116 | 0.461 | 0.555 | 0.399 | **0.609** | 0.586±0.028 |
| UWaveGestureL. | 0.570 | 0.482 | 0.710 | 0.713 | 0.582 | 0.419 | **0.758** | 0.557 | 0.559 | 0.728 | 0.742±0.014 |
| Average Acc. | 0.4680 | 0.4466 | 0.5150 | 0.5530 | 0.5132 | 0.3095 | 0.5348 | 0.4566 | 0.4943 | 0.7044 | **0.7154** |
| Average Rank | 7.0833 | 7.3333 | 5.8333 | 4.8333 | 6.4167 | 10.0833 | 5.5833 | 7.6667 | 7.2500 | 2.2917 | **1.6250** |

