# OpenReview forum: "CNN Kernels Can Be the Best Shapelets"
_ICLR.cc/2024/Conference — ICLR 2024 poster_

### Official Review · Reviewer_Lmj7 · 2023-10-26

**Soundness:** 2 fair
**Presentation:** 2 fair
**Contribution:** 2 fair
**Rating:** 5
**Confidence:** 4

**Summary:**

The authors introduce ShapeConv, an interpretable CNN layer whose kernels function as shapelets, designed for time series modeling in both supervised and unsupervised settings. They demonstrate that using the square norm in convolution, coupled with max pooling, is equivalent to computing the distance between a shapelet and a time series. Within this framework, a convolutional kernel essentially serves as a tunable shapelet. The authors also incorporate regularization to enforce similarity and diversity among shapelets, depending on whether the task is supervised (classification) or unsupervised (clustering). The methodology is validated through experiments on time series classification and clustering, using several competitor models and alternative implementations of ShapeConv for comparison. XAI is assessed via author-selected examples.

**Strengths:**

- The paper is generally well-structured and straightforward to follow.
- It establishes an interesting link between convolutional operations and the shapelet transform.
- The proposed methodology is versatile, applicable to both supervised and unsupervised tasks.

**Weaknesses:**

- The paper lacks a comprehensive review of related work, and the selection of competitor approaches for comparison is odd.
- Parts of the experimental section are unclear and require further clarification. The XAI evaluation is restricted to examples selected by the authors.
- There is no discussion or citation concerning code implementation.

**Questions:**

1. **Lack of Comprehensive Review of Related Work**:
   - The authors focus exclusively on optimization-based shapelet approaches. While space is limited, notable methods like Random Shapelet Forest and standard shapelet transform should not be omitted. Dictionary-based and interval-based approaches are also relevant and have achieved state-of-the-art performance in time series classification, yet they are not mentioned. Furthermore, the competitor models used in the experimental section are largely transformer-based or rely on embeddings, making for an unusual selection. I recommend that the authors thoroughly review relevant literature on time series classification, such as the paper by Ruiz et al. (2021) and models like ROCKET by Dempster et al. (2020).

2. **Ambiguities in the Experimental Section**:
    - Is the "Initialization" phase's cost included in the runtime?
    - In Table 1, why do the methods differ with respect to the cd plots?
    - Why is the evaluation limited to 25 UCR datasets, and what was the criteria for selection?
    - Several state-of-the-art methods like Rocket, CIF, ShapeletTransform, and MUSE are absent from the comparison.

3. **Limitations in XAI Evaluation**:
    - While the author-selected examples support the paper's claims, they do not suffice to demonstrate the superiority of ShapeConv in terms of shapelet quality. Additionally, pairing ShapeConv with MLP or SVM models does not provide sufficient interpretability. I suggest testing the approach with tree-based or linear models, or employing explainers such as SHAP to determine the importance of shapelets, especially in supervised tasks.

---

> ### Author Response · Authors · 2023-11-21
> **Response to Reviewer Lmj7 (Part 1)**
>
> Thanks for your valuable comments. Below we have addressed your questions and concerns point-by-point, and we have updated and added more clarifications to our manuscript accordingly.
>
> ### Weaknesses
>
> > The paper lacks a comprehensive review of related work, and the selection of competitor approaches for comparison is odd.
> >
> > Parts of the experimental section are unclear and require further clarification. The XAI evaluation is restricted to examples selected by the authors.
>
> We have addressed them in the Questions section.
>
> > There is no discussion or citation concerning code implementation.
>
> The code implementation is included in the supplementary material.
>
> ### Questions
>
> > Lack of Comprehensive Review of Related Work:
> >
> > - The authors focus exclusively on optimization-based shapelet approaches. While space is limited, notable methods like Random Shapelet Forest and standard shapelet transform should not be omitted. Dictionary-based and interval-based approaches are also relevant and have achieved state-of-the-art performance in time series classification, yet they are not mentioned. Furthermore, the competitor models used in the experimental section are largely transformer-based or rely on embeddings, making for an unusual selection. I recommend that the authors thoroughly review relevant literature on time series classification, such as the paper by Ruiz et al. (2021) and models like ROCKET by Dempster et al. (2020).
>
> Thank you for the suggestions.
>
> First, one kind of baseline models we have included in the Experiment section are *shapelet related methods, which includes not only optimization-based shapelet approaches*, but also the selection and generative based methods. The standard shapelet transform is denoted IGSVM in the table (Table F.1) and CD plot (Figure 3) according to the name in its paper. Besides, standard shapelet transform and its variants have also been mentioned in our related work like that: *Traditional practice is to search the raw datasets with some speed-up strategies, like paralleling computing (Chang et al., 2012), SAX transformation(Rakthanmanon & Keogh, 2013) and procedure simplification through newly designed measurements (Lines et al., 2012; Guillaume et al., 2022; Zakaria et al., 2012).*
>
> Second, other two kinds of baselines are common deep learning methods (MLP, CNN, RNN) and state-of-the-art methods (TS2Vec, TST, DTW, etc), which covers various types of neural networks.
>
> Third, we have added ROCKET as a baseline in the UCR dataset comparison upon your request. However, we also want to mention that ROCKET is already compared with some of our baselines, like TST and OS-CNN, in their papers. Since we are comparing with the state-of-the-art methods in recent two years, and many previous methods like HIVE-COTE and BOSS are already compared in our baselines' papers, we believe that superior performance over these state-of-the-art methods can well demonstrate the effectiveness of our model.
>
> Finally, we have also updated our related work section by introducing non-deep-learning approaches like the dictionary-based method MUSE and the ensemble method HIVE-COTE.
>
> > Is the "Initialization" phase's cost included in the runtime?
>
> Yes, it is included. We have clarified this in Table 1.
>
> > In Table 1, why do the methods differ with respect to the cd plots?
> >
> > Why is the evaluation limited to 25 UCR datasets, and what was the criteria for selection?
>
> In the CD plot (Figure 3), ShapeConv is compared with shapelet-based methods on 25 UCR datasets, and full results here are shown in Table F.1 in Appendix. In Table 1, ShapeConv is compared with state-of-the-art time-series classification models on 125 UCR and 29 UEA datasets, and the full results are shown in Table F.3 and F.4. Methods are different since they are two different experiments.
>
> We use the subsets (25 UCR datasets) in the CD plot because the baseline (ADSN) only reported results on 25 UCR dataset, and the results of baselines are taken directly from the respective papers. Full results of ShapeConv and state-of-the-art classification methods on all 128 UCR datasets are shown in Table F.3 in Appendix.

---

> > ### Author Response · Authors · 2023-11-21
> > **Response to Reviewer Lmj7 (Part 2)**
> >
> > > Several state-of-the-art methods like Rocket, CIF, ShapeletTransform, and MUSE are absent from the comparison.
> >
> > We have added Rocket as a baseline in Table 1 and F.3 upon your request, and the ShapeletTransform (namely IGSVM) is already compared in the paper. We found the results of CIF and MUSE in [1], and the average accuracy on the 109 subset of UCR is as follows:
> >
> > |  | ShapeConv | CIF | MUSE |
> > | - | - | - | - |
> > | Average Accuracy  | 0.856 | 0.846 | 0.845 |
> >
> > All these results well demonstrate the effectiveness of ShapeConv.
> >
> > Besides, we would like to emphasize that ShapeConv not targets on beating all state-of-the-art methods. It offers great benefits on interpretability and is able to seamlessly integrate with other deep learning models.
> >
> > [1] Tan, C.W., et al. MultiRocket: multiple pooling operators and transformations for fast and effective time series classification. Data Min Knowl Disc 36, 1623–1646 (2022).
> >
> > > Limitations in XAI Evaluation:
> > > - While the author-selected examples support the paper's claims, they do not suffice to demonstrate the superiority of ShapeConv in terms of shapelet quality. Additionally, pairing ShapeConv with MLP or SVM models does not provide sufficient interpretability. I suggest testing the approach with tree-based or linear models, or employing explainers such as SHAP to determine the importance of shapelets, especially in supervised tasks.
> >
> > We thank the reviewer for you insightful comments. The quality of shapelet, as well the quality of all the other interpretable methods, should be judge from two different perspectives: the effectiveness and the interpretability.
> >
> > As for the effectiveness, we tested our model on a bunch of datasets compared to several baselines, and the experiment results have demonstrated the superiority in Figure 3, Table 1, Table D.1, Table F.1~F.6.
> >
> > As for the interpretability, we have to admit that the evalution itself is a very tough research question in the interpretable time-series modeling task. However, we still designed the following approaches to verify that our methods are truly interpretable:
> > 1. Figure 4, 5, 6, C.1 compared the shapelets learned by ShapeConv and the state-of-the-art learning-based shapelet baseline, LTS, and our shapelets are apparently fit better with the data.
> > 2. Appendix A provides a detailed explanation of the learned shapelets in a human-perceivable way on a real-world dataset, GunPoint, to demonstrate how these shapelets can be interpreted.
> >
> > And we argue that pairing ShapeConv with MLP or SVM, and even deep models demonstrated in Appendix D, is designed to obtain better effectiveness and shouldn't harm the interpretabilty. This is because ShapeConv serves as a feature extractor conducting shapelet transform which directly engaged with the raw time series, and is sensitive to the difference of shapes. This idea first comes as shapelet transform [2] and has become a common practice followed by the following research works. As a matter of fact, all the shapelets visualized and analyzed are trained in an end-to-end manner and we didn't see any negative impact for interpretability.
> >
> > Moreover, we added the following contents to the revised manuscript:
> > 1. In Appendix D., we further investigate the interpretability of shapelets learned by ShapeConv for the real-world EEG classification task, and we excitedly found that some shapelets are very similar to textbook waveforms which have already been recognized in clinical practice.
> > 2. In Appendix E.1, we add more visualization examples of learned shapelets.
> > 3. In Appendix E.2, we add visualizations of the SHAP values for ShapeConv and its variant without the shape regularizer (essentially a CNN with our initializations and diversity loss). The significant regions labeled by the SHAP value in the ablation overlaps with the determining regions of the ShapeConv, verifying the quality of the shapelets. Furthermore, ShapeConv excels in not only identifying the significant regions of the input, but also captures the shape of the discriminative subsequence. This capability sets it apart from traditional explainability methods like SHAP, which primarily focus on pinpointing the locations of these areas.
> >
> > Please kindly refer to the revised version of our paper for details and figures.
> >
> > [2] Jason L., et al. A shapelet transform for time series classification. In Proceedings of the 18th ACM SIGKDD international conference on Knowledge discovery and data mining, pp. 289–297, 2012.

---

### Official Review · Reviewer_JpzU · 2023-10-30

**Soundness:** 2 fair
**Presentation:** 3 good
**Contribution:** 3 good
**Rating:** 5
**Confidence:** 5

**Summary:**

The paper proposes a modified convolutional layer for time series analysis inspired by the Shapelet distance that is widely used in the domain.
This new layer is then used at the core of neural networks for both supervised and unsupervised tasks.
A regularization term for the task-specific losses is designed that enforces learned kernels to (i) look like actual subseries from the training set and (ii) form a diverse set.

**Strengths:**

The paper is very well written and the motivation for the method is clear.
The experimental validation is quite thorough and it is nice to showcase that the method can be used for both supervised and unsupervised learning.
In terms of the method, while the idea of having a Shapelet layer included in a neural network is not novel, both the initialization scheme and the regularization terms included in the loss lead to improvement on the performance of the resulting models.

**Weaknesses:**

In the abstract (it is also said in the introduction in other words), it is stated that:

>  In this paper, we demonstrate that shapelets are essentially equivalent to a specific type of CNN kernel with a squared norm and pooling

In fact, this demonstration is not novel, it is for example stated in (Lods et al. 2017) (that is cited in the paper).
However, it seems that here, the proof aims at more rigor, but Theorem 3.1 is not successful in this regard since it completely disregards the fact that the bias term in convolution is independent of the input, which is not the case in the $-\mathcal{N}(s_i, X_{j:j+l_s-1})$ term. (Also, as a side note, in Theorem 3.1, Squared Euclidean distance is used, not Euclidean distance as stated.)

Moreover, the review of the Related Work is very succinct and a more thorough presentation of competing interpretable Shapelet-based methods would have been a plus.
Similarly, a more detailed comparison of the interpretability of the ShapeConv model with those baselines is required to fully assess interpretability:
* Only toy examples are presented (eg. Fig 4: 2 shapelets), what does it give when training with a large amount of shapelets?
* Also, providing visualization for a large number of datasets instead of only GunPoint+Herring+ECG200 would be a real plus

**Questions:**

Apart from the questions/suggestions related to the evaluation of interpretability, I have a few remarks/questions that are listed below:

* If you took your your ShapeConv model (with exact same initialization, regularization terms, etc.) and changed the ShapeConv layer with a convolutional one, what would you get in terms of performance? This experiment is required to fully assess if the norm terms a really helpful
* In terms of evaluation:
    * How are baseline model hyperparameters tuned (and which parameters are tuned)?
    * How do you pick the datasets for the subsets (25 datasets for supervised learning and 36 datasets for unsupervised learning)?
    * If the goal is to compare to state-of-the-art methods, other competitors should be included in the comparison (eg. ROCKET, COTE & variants, ...)

Below are some minor remarks/questions:
* In Section 1, you write:
    >  they are more likely to overfit when the signal-to-noise ratio is relatively large
    * Don't you mean "is relatively low"?
* Initialization
    * Have you assessed how important it was to use supervised information at initialization?
    * Have you tried simpler approaches (eg. kmeans++ on randomly selected subsequences of adequate length)?
* If ShapeConv is faster than LTS, it is probably more an artifact of the implementation since the overall complexity of ShapeConv is probably higher than that of LTS (similar local representation extracted, but ShapeConv have additional loss terms that induce more computations)
* Presentation
    * Unsupervised learning: it is unclear from the presentation in Section 3.4 which clustering method is used on top of the features extracted from ShapeConv. This is detailed in Section 4.2, but should be explained in Section 3.4 imho

---

> ### Author Response · Authors · 2023-11-21
> **Response to Reviewer JpzU (Part 1)**
>
> Thanks for your valuable comments. Below we have addressed your questions and concerns point-by-point, and we have updated and added more clarifications to our manuscript accordingly.
>
> ### Weaknesses
>
> >In the abstract (it is also said in the introduction in other words), it is stated that:
> >
> >> In this paper, we demonstrate that shapelets are essentially equivalent to a specific type of CNN kernel with a squared norm and pooling
> >
> > In fact, this demonstration is not novel, it is for example stated in (Lods et al. 2017) (that is cited in the paper). However, it seems that here, the proof aims at more rigor, but Theorem 3.1 is not successful in this regard since it completely disregards the fact that the bias term in convolution is independent of the input, which is not the case in the $-\mathcal{N}(\mathbf{s_i}, \mathbf{X}_{j:j+l_s-1})$ term.
>
> We would like to emphasize that the bias term in convolution has no relation with the $-\mathcal{N}(\mathbf{s\_i}, \mathbf{X}\_{j:j+l\_s-1})$ term. The bias term, as we have explained in the end of Sec. 3.1, can be optionally added after the features are extrated with the shapelets (or CNN kernels). And the $-\mathcal{N}(\mathbf{s_i}, \mathbf{X}_{j:j+l_s-1})$ term is an additional squared L2 norm term added in ShapeConv besides the cross-correlation operator in CNN. The equivalence cannot be realized without this additional squared L2 norm. Therefore, the bias term do not influence the proof of equivalence, so that Theorem 3.1 is successful in the rigor equivalence between CNNs and shapelets.
>
> > Also, as a side note, in Theorem 3.1, Squared Euclidean distance is used, not Euclidean distance as stated.
>
> Thanks for pointing out. We have revised our manuscript.
>
>
> > Moreover, the review of the Related Work is very succinct and a more thorough presentation of competing interpretable Shapelet-based methods would have been a plus. Similarly, a more detailed comparison of the interpretability of the ShapeConv model with those baselines is required to fully assess interpretability:
> > - Only toy examples are presented (eg. Fig 4: 2 shapelets), what does it give when training with a large amount of shapelets?
> > - Also, providing visualization for a large number of datasets instead of only GunPoint+Herring+ECG200 would be a real plus
>
> For the visualizations, we did not choose "toy examples" - the ShapeConv in Fig 4 could achieve 0.75 accuracy on the Herring dataset with 2 shapelets. When the number of shapelets are large, the model tends to overfit for these simple datasets. As for examples with large number of shapelets, we provided a real-world application of ShapeConv in EEG (Appendix D) where 2,688 shapelets are trained. Two of them are visualized in Figure D.1, and they matched nicely with the EEG examples from textbook.
>
> We also provided more visualizations in the appendix E, showing learnt shapelets in 4 additional datasets from UCR: DodgerLoopWeekend, SonyAIBORobotSurface1, ItalyPowerDemand, BME. ShapeConv captures the determinative regions in all of them.
>
> In terms of interpretability, ShapeConv is generally shapelet-based method, so its interpretability inherits from shapelet. We have compared the interpretation results of ShapeConv with the other shapelet-based method (i.e., LTS) via visualizations, and results shows that ShapeConv is better at capturing discriminative sub-sequences, thus having better interpretability.
>
>
> ### Questions
>
> > If you took your your ShapeConv model (with exact same initialization, regularization terms, etc.) and changed the ShapeConv layer with a convolutional one, what would you get in terms of performance? This experiment is required to fully assess if the norm terms a really helpful
>
> We anticipate your proposed ablation is to drop the squared $L_2$ norm term $\mathcal{N}(\mathbf{s_i}, \mathbf{X'})$ from Equation (4) in the loss. However, then the shape regularizer in the loss would become
> $$
> \mathcal{R}\_{shape}=\frac{1}{n_{out}}\sum_{i=1}^{n_{out}}  \min_{\mathbf{x}\in\hat{\mathbf{X}}_i} -2Y(\mathbf{s_i},\mathbf{x})
> $$
> where $Y(\mathbf{s_i},\mathbf{x})$ denotes the cross-correlation between kernel $\mathbf{s_i}$ and subsequence $\mathbf{x}$. Minimizing this regularizer would make the loss diverge and goes to negative infinity. This is because without the norm terms, the regularizer could not be interpreted as a distance and is not always nonnegative.
>
> If we do not use the shape regularizer, the formulation would become similar with the CNN baseline. We tested the ShapeConv without Shape Loss (by setting the $\lambda_{shape}$ in Equation (8) to zero). The results for the GunPoint dataset is illustrated in appendix E.2. We found that its kernel is not interpretable and it performs worse than ShapeConv (it constantly overfits the training set). The result performance is similar with the CNN baseline. We did not include the full results in the paper due to space constraints.

---

> > ### Author Response · Authors · 2023-11-21
> > **Response to Reviewer JpzU (Part 2)**
> >
> > > How are baseline model hyperparameters tuned (and which parameters are tuned)?
> >
> > The results of the baseline models are taken directly from their papers. We have clarified this in the paper.
> >
> > > How do you pick the datasets for the subsets (25 datasets for supervised learning and 36 datasets for unsupervised learning)?
> >
> > We choose these subsets following the baseline methods (ADSN for supervised and AutoShape for unsupervised) which only reported the results on these dataset. We have clarified this in the paper.
> >
> > > If the goal is to compare to state-of-the-art methods, other competitors should be included in the comparison (eg. ROCKET, COTE & variants, ...)
> >
> > We have added comparison with ROCKET in Appendix Table F.3. COTE is already compared with the baseline method ADSN in their paper. We have not included these traditional methods in the main paper, but lean towards state-of-the-art deep-learning-based methods for the following reasons: (1) DL-based methods generally have more competitive performance; (2) ShapeConv can be generally regarded as DL-based methods, so both of them can be paralleled on GPU, achieving much faster runtime than traditional methods on CPU; (3) In terms of model expressiveness, comparing ShapeConv with DL-based methods is more reasonable, since both of them are neural network models. (3) Since our model is written in the PyTorch framework and could automatic differentiate, our method can act as a feature extractor and seamlessly integrate with other DL-based methods.
> >
> > > Don't you mean "is relatively low"?
> >
> > Thank you for pointing out. We have fixed the typo.
> >
> > > Initialization
> > > - Have you assessed how important it was to use supervised information at initialization?
> > > - Have you tried simpler approaches (eg. kmeans++ on randomly selected subsequences of adequate length)?
> >
> > Yes, we have done these two studies. First, we have the random initialization ablation in Figure 3 and Table F.1. Random initialization leads to large performance degeneration, indicating the importance of our designed initialization methods and supervised information at initialization.
> >
> > Besides random initialization, we have also attempted many different ways. Your proposed method is similar with our unsupervised initialization approach - the only difference is that we first divide the input sequence into different regions and then do the sampling and k-means. Moreover, we have also tested your proposed method before and it was not as good as our current method. For supervised learning, such simpler approaches do not incorporate important supervised information. For unsupervised learning, pre-dividing according to regions makes clustering easier and contributes to better initialization. Therefore, we end up choosing the best tested approach as our current method.
> >
> > > If ShapeConv is faster than LTS, it is probably more an artifact of the implementation since the overall complexity of ShapeConv is probably higher than that of LTS (similar local representation extracted, but ShapeConv have additional loss terms that induce more computations)
> >
> > Our method is faster than LTS is because our model is paralleled on GPU, while LTS is implemented on CPU (we tested against the LTS implementation in [tslearn](https://github.com/tslearn-team/tslearn/)). Additional loss terms are negligible in terms of computation during runtime compared to the convolution operation. While the theoretical computational complexity is almost the same, being able to compute forward pass and gradient in parallel makes ShapeConv much faster and better at learning shapelets.
> >
> >
> > > Unsupervised learning: it is unclear from the presentation in Section 3.4 which clustering method is used on top of the features extracted from ShapeConv. This is detailed in Section 4.2, but should be explained in Section 3.4 imho
> >
> > Thank you for pointing out. We update Section 3.4 by adding the clustering method (i.e., K-means).

---

> > > ### Comment · Reviewer_JpzU · 2023-11-22
> > > **Answer to part 2**
> > >
> > > Thank you for this feedback (part 2), it makes things clearer for me.

---

> > ### Comment · Reviewer_JpzU · 2023-11-22
> > **Answer to part 1**
> >
> > I still don't understand what Theorem 3.1 means by "equivalence".
> >
> > In order to prove equivalence between CNNs and Shapelets, you need to exhibit that for a given set of shapelets, the same transformation can be achieved using a given kernel and vice versa, which is not done. Once again, I do not see how the $\||X\||^2_2$ term can be obtained at the output of a convolution operation.
> >
> > What you prove is that Shapelet transform is identical to your ShapeConv (which is not a standard convolution).
> > In other words, ShapeConv is just a re-writing of the shapelet transform, and is not a convolutional layer.
> >
> > This relates to the question on complexity: as stated in your answer, the only difference between your ShapeConv and tslearn's implementation of LTS is that you run yours on GPU, but `tslearn` uses the same tricks as you do to compute the shapelet transform it seems: https://github.com/tslearn-team/tslearn/blob/09441abdeb3056a47615a299cbd41499def4dd46/tslearn/shapelets/shapelets.py#L141 hence could probably be run on GPU with the same time complexity.

---

> > > ### Author Response · Authors · 2023-11-22
> > > **Equivalence between CNN kernel and Shapelet**
> > >
> > > Thank you for your valuable feedback. We address you points as follows:
> > >
> > > **The equivalence between CNN kernel and Shapelet**: As we stated in our paper, the equivalence is established between shapelets and the kernels of ShapeConv, a specific type of CNN. We now follow the way you suggested to roughly prove the equivalence. Setting the kernel of the ShapeConv as predefined shapelets would yield the same results as the standard shapelet-transform, since ShapeConv involves the additional $-\mathcal{N}(\mathbf{s_i}, \mathbf{X}_{j:j+l_s-1})$ term according to Theorem 3.1, and we also well handle the non-linear activation and optional bias term, so that the $||X||^2_2$ term can be obtained at the output of ShapeConv. Conversely, kernels learned from ShapeConv can be utilized as shapelets in all shapelet-based methods. Our claim has not been extended to a broad equivalence between general CNNs and shapelets.
> > >
> > > Based on this equivalence, we want to emphasize that implementing the equivalence in practice is complex and demands careful design choices, such as specific regularizations for similarity and diversity, and tailored initialization approaches for both supervised and unsupervised learning. We also demonstrate how human knowledge can be integrated through initialization, particularly valuable in real-world contexts like clinical applications. We argue that these are also main contributions of our paper which should not be omitted.
> > >
> > > **Comparison to LTS**: We indeed rewrite the shapelet-transform into two parts: a vanilla CNN layer efficiently handled by deep learning frameworks like PyTorch, and the normalizing terms, also computed using these frameworks for parallel processing and automatic differentiation. Consequently, our implementation benefits from the optimized convolution layers in deep learning frameworks (e.g., PyTorch's 1D Convolution layer leveraging cuDNN's optimized CUDA functions), rendering ShapeConv substantially faster than LTS. As we mentioned in the paper, this gain is particularly evident when dealing with long shapelets, as convolution layers handle gradients more effectively. Furthermore, employing CNNs offers additional benefits, such as adjustable stride and enabling the usage of dilated convolutions, while these features are not available in LTS. All these benefits are based on the equivalence and our implementation, which are not some separate trivial parts.

---

### Official Review · Reviewer_nftB · 2023-10-31

**Soundness:** 3 good
**Presentation:** 3 good
**Contribution:** 3 good
**Rating:** 8
**Confidence:** 5

**Summary:**

The paper combines Shapelets and CNNs

**Strengths:**

Good empirical results.
Tests on many datasets (although, just download and test, no new datasets)

**Weaknesses:**

I appreciate the accuracy improvements, while small, are probably real.

However, I have no confidence in any of the claims of interpretability and explainability.
You made no effort to obtain the original herring images or gun-point videos. You explanations here are "just-so" stories. [v].

If you wanted to make convincing claims here, you could obtain the herring images, modify them to add / remove effects, then see how this affects the shapelets. Or reenact the gun-point video, and  modify the protocol to add / remove effects, or...

I do understand that most people in this space are too lazy to go beyond downloading the UCR datasets. But if that is all you do, it seems like you should temper your claims about interpretability and explainability.





“In the realm of machine learning, interpretable time-series modeling stands as a pivotal endeavor, striving to encode sequences and forecast in a manner that resonates with human comprehension”
This (and the rest of the paper) read like flowery language [a].


In fig 1, can you move the legend away from the data?


“is evaluated on the 25 UCR” Did you mean “125” or “25”?


“Figure 5: Shaplets learned” typo (Shapelets)


“It is evident that the shapelet learned by ShapeConv captures the distinguishing features of the class effectively”  Evident to whom? You should argue that the blue shapelets correctly represents the actors hand having to hover over the gun holster, then reach down to the gun, then draw the gun.


“clustering task using 36 UCR univariate”
Why 36? Why this particular 36?



“In response to the first RQ, we observe that ShapeConv’s shapelets (Figure 4 (a)) cover all turning points in the time series, where the two classes differ the most, while LTS’s shapelets (Figure 4 (b)) do not cover the targeted regions.”
This evaluation is tautological. If  “turning points” are the best places for shapelets, then we don’t need any search for shapelets at all.



“In contrast, when using human initialization,..”
Hmm, it is a bit tricky to claim results based on human initialization. Which humans, how trained are they in the system, how are they briefed. In my mind, that is a separate “human in the loop” paper.


However, despite ingenious, the performance (missing a word?)
However, despite ingenious suggestions, the performance


gun out of the gun pocket (holster)


“while data from the “finger” class don’t.”
“while data from the “finger” class do not.”  (avoid contractions in scientific writing)


This illustrate how
This illustrates how

In table E.5, why four significant digits? This is spurious  accuracy.

In table E.5 and elsewhere, you report the average accuracy.  This is meaningless for datasets of different sizes, class skews, number of classes, default rates etc. To be clear, it is not a flawed metric, it is just meaningless.


[a] https://www.latimes.com/socal/daily-pilot/opinion/story/2022-05-03/a-word-please-flowery-writing-can-turn-off-readers

[v] https://en.wikipedia.org/wiki/Just-so_story

**Questions:**

NA

---

> ### Author Response · Authors · 2023-11-21
> **Response to Reviewer nftB**
>
> Thanks for your valuable comments. Below we have addressed your questions and concerns point-by-point, and we have updated and added more clarifications to our manuscript accordingly.
>
> ### Weaknesses
>
> > However, I have no confidence in any of the claims of interpretability and explainability. You made no effort to obtain ... go beyond downloading the UCR datasets. But if that is all you do, it seems like you should temper your claims about interpretability and explainability.
>
> Thank you for the suggestions. We tried to access the original data of the GunPoint and Herring dataset, but it is not available. Nevertheless, to further support our claims about interpretability, we have added a real-world example usage of ShapeConv in EEG when stacked with deep models. Breifly speaking, we have excitedly discovered that a few shapelets are highly similar to waveforms in clinical textbooks. By aligning shapelets with data and conduct shapelet transform, we see clear separatable clusters. The overall experiments in Appendix D demonstrate the possibility of ShapeConv, when stacking with deep models, can benefit medical practitioners in practice by not only offering an accurate judgement, but also pointing out the area of interests with respect to their expert knowledge. **Please kindly refer to the revised version of our manuscript, Appendix D, for figures and more details.**
>
> > “In the realm of machine learning, interpretable time-series modeling stands as a pivotal endeavor, striving to encode sequences and forecast in a manner that resonates with human comprehension” This (and the rest of the paper) read like flowery language [a].
>
> Thanks for the suggestion. We will go through the paper and try to make the sstatements simplified and less confusing in the final version.
>
> > In fig 1, can you move the legend away from the data?
>
> Thank you for pointing out. We have moved the legend away from the data.
>
> > “is evaluated on the 25 UCR” Did you mean “125” or “25”?
>
> Here the 25 UCR dataset is the subset of the 128 UCR dataset. We use this subset because the baseline method (ADSN) only reported this subset. We have clarified this in the paper.
>
> > “Figure 5: Shaplets learned” typo (Shapelets)
>
> Thank you for pointing out. We have fixed the typo.
>
> > “It is evident that the shapelet learned by ShapeConv captures the distinguishing features of the class effectively” Evident to whom? You should argue that the blue shapelets correctly represents the actors hand having to hover over the gun holster, then reach down to the gun, then draw the gun.
>
> Thank you for the suggestions. We have added more details in the paper.
>
> > “clustering task using 36 UCR univariate” Why 36? Why this particular 36?
>
> We use the 36 UCR dataset because the baseline method (AutoShape) only reported this subset. We have clarified this in the paper.
>
> > “In response to the first RQ, we observe that ShapeConv’s shapelets (Figure 4 (a)) cover all turning points in the time series, where the two classes differ the most, while LTS’s shapelets (Figure 4 (b)) do not cover the targeted regions.” This evaluation is tautological. If “turning points” are the best places for shapelets, then we don’t need any search for shapelets at all.
>
> In this example, the turning points are the best places for shapelets. However, we agree that this is not always the case. We have clarified this in the paper.
>
> > “In contrast, when using human initialization,..” Hmm, it is a bit tricky to claim results based on human initialization. Which humans, how trained are they in the system, how are they briefed. In my mind, that is a separate “human in the loop” paper.
>
> We agree that this is tricky to claim. Our goal is to show that ShapeConv has the ability to use human knowledge as prior, which is not possible in other shapelet-based models.
>
> > However, despite ingenious, the performance (missing a word?) However, despite ingenious suggestions, the performance
> >
> > gun out of the gun pocket (holster)
> >
> > “while data from the “finger” class don’t.” “while data from the “finger” class do not.” (avoid contractions in scientific writing)
> >
> > This illustrate how This illustrates how
>
> Thank you for carefully reading the paper and pointing out these typos. We have fixed the typos.
>
> > In table E.5, why four significant digits? This is spurious accuracy.
>
> We use four significant digits because the results are taken directly from the respective papers. We have revised this in the paper to make it consistent.
>
> > In table E.5 and elsewhere, you report the average accuracy. This is meaningless for datasets of different sizes, class skews, number of classes, default rates etc. To be clear, it is not a flawed metric, it is just meaningless.
>
> We agree that the average accuracy is meaningless for different datasets. However, the community has been using this metric for a long time. We have to report this metric to compare with the previous methods.

---

> > ### Comment · Reviewer_nftB · 2023-11-21
> >
> > You say "We tried to access the original data of the GunPoint and Herring dataset, but it is not available".
> >
> > How did you try to "access the original data of the GunPoint"?
> >
> > Two years ago my student asked the original auhours for "the original data of the GunPoint", They not only gave her the original data, they gave her the orginal video and photographs from the recording session, and they answered all her questions about it etc.
> > It is very surprising that they were so helpful to one researcher, but not to another.

---

> > > ### Author Response · Authors · 2023-11-22
> > > **Interpretation of the GunPoint Dataset**
> > >
> > > Thanks for sharing these experiences with us as well as the courage to communicate with authors! We searched on Web and papers for public accessing before since we thought that this dataset was released a long time ago so reaching out the authors might bother them a little bit (and we are too shy to communicate). But, we have e-mailed the authors after seeing your comments, and have discussed with them for several rounds.
> > >
> > > We have found more supporting evidences from the original data shared by the authors. Specifically, photographs are frames in the video, and every frame highlights the position of hand. From the gun videos, we have identified corresponding frames of shapes distinguished by ShapeConv (shown in Figure A.3 in Appendix). These frames verify our claim that "these two shapes corresponds to the motion of a hand pulling a gun out of the holster and put it back".
> > >
> > > As for reenacting the gun-point video, and modifying the protocol to add / remove effects, we currently have no concrete idea on how to do these. Even though now we have frames from the original video, it seems that we still cannot intervene the video generation process as we wish. To be specific, we are not sure that how we can set a specific experiment condition, and then change the video based on the original ones, unless we reenact the gun-point video. However, such reenacting requires rigorous designs on the data collection pipeline and appropriate scene setup to mimic a real-world scenario, which is hard to finish during this rebuttal phase. We also feel it hard to properly design the reenacting experiments based on the existing findings, i.e., the correspondence between the original video frame and our discovered shapelets.
> > >
> > > As for the Herring dataset, since we did not include detailed interpretation as GunPoint dataset in our paper, we plan not to add the similar analysis to the revision. But we'll (and we are) actively looking for the resources to verify our effectiveness of shapelets.
> > >
> > > Besides, before collecting new dataset, we would like to bring your attention to our EEG experiments in Appendix D. EEG signals are time-series data collected under real-world scenarios, and the dataset is much larger than UCR datasets. We show that some interpretable kernels have very similar shape with examples in the textbook, matching exactly with knowledge of human experts. We believe that this is a strong proof for the effectiveness of ShapeConv.
> > >
> > > If you have more suggestions on this problem, please let us know. We are very happy and open to have further discussion and will take your suggestions into consideration even after the reviewing period.

---

### Official Review · Reviewer_JCHN · 2023-10-31

**Soundness:** 3 good
**Presentation:** 3 good
**Contribution:** 3 good
**Rating:** 6
**Confidence:** 3

**Summary:**

The paper bridges the divide between traditional shapelets and modern deep learning methods in time-series modeling. Shapelets, while interpretable, face efficiency issues; deep learning models offer performance but lack interpretability. The proposed ShapeConv melds these approaches, using a CNN layer with its kernel functioning as shapelets. This layer is both interpretable and efficient, achieving state-of-the-art results in experiments. The introduction of shaping regularization and human knowledge further enhances its performance and interpretability.

**Strengths:**

1. This paper theoretically establishes an equivalence between traditional Shapelets and using a convolutional layer to derive similar features. It’s a fresh perspective in utilizing shapelets in combination with deep learning methods and structure.
2. The comprehensive experiments empirically demonstrate the superior performance of ShapeConv, in both classification as well as clustering tasks.
3. The paper is well-written and easy to understand.

**Weaknesses:**

1. An analysis of the computational complexity and resource requirements of ShapeConv could make the paper more comprehensive.
2. Though the model's performance is promising, concerns may arise regarding the complexity of implementing ShapeConv compared to other traditional or deep learning models.

**Questions:**

This study opted for a combination of CNN and Shapelets to enhance interpretability while also boosting performance. For time series classification tasks, why not choose the stronger baseline models for research, such as RNN or Transformer?

---

> ### Author Response · Authors · 2023-11-21
> **Response to Reviewer JCHN**
>
> Thanks for your valuable comments. Below we have addressed your questions and concerns point-by-point, and we have updated and added more clarifications to our manuscript accordingly.
>
> ### Weaknesses
>
> > An analysis of the computational complexity and resource requirements of ShapeConv could make the paper more comprehensive.
>
> The ShapeConv has the same computational complexity and resource requirements as the 1D convolution. The only difference is that ShapeConv has an additional regularization term and additional losses. These terms are negligible compared to the convolution operation. The total computational complexity of ShapeConv for a univariate time series is $O(l_x \times l_s \times n_{out})$, where $l_x$ is the length of the input time series; $l_s$ is the length of the shapelet, and $n_{out}$ is the number of shapelets. The advantage of ShapeConv is that is could be paralleled in GPU, making is much faster than traditional methods on CPU (2000x faster than LTS).
>
> > Though the model's performance is promising, concerns may arise regarding the complexity of implementing ShapeConv compared to other traditional or deep learning models.
>
> ShapeConv is implemented as a Module in PyTorch similar to a common CNN layer, which can be easily integrated as a layer in any deep learning model. The implementation of ShapeConv is available at the supplementary material. Besides, the implementation of ShapeConv itself is simple and intuitive, which contains the convolution layer and some additional regularizers, aligning with Eq. (4) in Theorem 3.1.
>
>
> ### Questions
>
> > This study opted for a combination of CNN and Shapelets to enhance interpretability while also boosting performance. For time series classification tasks, why not choose the stronger baseline models for research, such as RNN or Transformer?
>
> RNN and Transformer-based methods are already included in our baselines. Table F.2 shows results of ShapeConv and various RNN-based methods, and we also compared with Transformer-based model, TST [1], in Table 1 and appendix Table F.3. ShapeConv outperforms these methods on most of the tasks. These methods are not listed in the main paper since we only include most related and competitive methods due to the limited spaces.
>
> [1] George Z., et al. A transformer-based framework for multivariate time series representation learning. In Proceedings of the 27th ACM SIGKDD Conference on Knowledge Discovery & Data Mining, pp. 2114–2124, 2021.

---

### Official Review · Reviewer_VPQr · 2023-11-07

**Soundness:** 2 fair
**Presentation:** 3 good
**Contribution:** 1 poor
**Rating:** 5
**Confidence:** 3

**Summary:**

This article deals with the classification of time series. The authors describe the equivalence between a particular approach, shapelets, and convolutional layers. They provide several losses to enforce the diversity of learned shapelets and closeness to original data, as well as intuitive initialization methods. The proposed approach is compared to several algorithms in a thorough experimental study.

**Strengths:**

The article describes the methodology well, and, to the best of my knowledge, the proposed initializations and losses are novel in the context of shapelets. The experiment study is extensive (with one caveat, see below) and convincing.

**Weaknesses:**

- The main contribution is based on Theorem 3.1, which shows that the shapelet transform is somewhat equivalent to a convolution layer followed by a max pooling operation. However, this fact has been observed previously to provide accelerated shapelet transform: the authors of [1] show that computing the distance profile ($dist(\mathbf{s}, \mathbf{x})$ for a given sequence $\mathbf{s}$ and all subsequences $\mathbf{x}$ of $\mathbf{X}$) is equivalent to a convolution.

- An extensive review of time series classification algorithms exists on the same data sets, see [2] and more recently but unpublished [3]. None of the algorithms referenced in [2, 3] are compared to ShapeConv. The authors should at least compare themselves to the best-performing algorithms of the state-of-the-art.

- (Minor comment.) It is considered bad practice to start sentences with mathematical symbols.

[1] Yeh, C. C. M., Zhu, Y., Ulanova, L., Begum, N., Ding, Y., Dau, H. A., Zimmerman, Z., Silva, D. F., Mueen, A., & Keogh, E. (2016). Matrix Profile I: All Pairs Similarity Joins for Time Series: A Unifying View that Includes Motifs, Discords and Shapelets. Proceedings of the IEEE International Conference on Data Mining (ICDM), 1317–1322. https://doi.org/10.1007/s10618-017-0519-9

[2] Bagnall, A., Lines, J., Bostrom, A., Large, J., & Keogh, E. (2017). The great time series classification bake off: a review and experimental evaluation of recent algorithmic advances. Data Mining and Knowledge Discovery, 31(3). https://doi.org/10.1007/s10618-016-0483-9

[3] Middlehurst, M., Schäfer, P., & Bagnall, A. (2023). Bake off redux: a review and experimental evaluation of recent time series classification algorithms. ArXiv. http://arxiv.org/abs/2304.13029

**Questions:**

In addition to addressing my comments about Theorem 3.1 and the comparison to the state-of-the-art, I have one question:
- Convolutional layers are meant to be stacked. Unless I am mistaken, in the experiments, there is only one ShapeConv layer. Would the interpretability of ShapeConv remain if there are several layers?

---

> ### Author Response · Authors · 2023-11-21
> **Response to Reviewer VPQr (Part 1)**
>
> Thanks for your valuable comments. Below we have addressed your questions and concerns point-by-point, and we have updated and added more clarifications to our manuscript accordingly.
>
> ### Weaknesses
>
> > The main contribution is based on Theorem 3.1, ... is equivalent to a convolution.
>
> We argue that our work is definitely different from the work in [1] despite some faint connections. The authors of [1] discuss the connection between z-normalized Euclidean distance and dot products so as to utilize convolution techniques such as the fast Fourier transform (FFT) to achieve speed-ups in searching similar subsequences. The method are mentioned to "*suggest*" candidate shapelets as *indicators*, and barely limited to sub-sequences of the original time series which cannot meet the needs of deep learning. We, to be noticed, rigorously prove the equivalence between a certain neural convolutional layer (not just convolution) and the originally defined shapelet in order to propose a newly interpretable and learnable neural kernel for deep learning.
>
> In fact, we have properly cited the papers of [2] and [3] (also discuss the relationship between convolution and shapelet) which are more relevant and compared them with our work to highlight our contribution of demonstrating the strict equivalence between shapelet transform and a certain learnable neural operator. With GPUs’ built-in parallel computation of for deep learning, our ShapeConv is naturally high-speed and does not need extra algorithms specifically designed for acceleration. Besides, we also put forward the diversity regularization and incorporate human-knowledge-based initialization to enhance the interpretability of our proposed method as non-negligible contributions.
>
> [1] Yeh, C. C. M., et al. (2016). Matrix Profile I: All Pairs Similarity Joins for Time Series: A Unifying View that Includes Motifs, Discords and Shapelets. Proceedings of the IEEE International Conference on Data Mining (ICDM), 1317–1322.
>
> [2] Arnaud L., et al. Learning dtw-preserving shapelets. In Advances in Intelligent Data Analysis XVI: 16th International Symposium, IDA 2017, London, UK, October 26–28, 2017, Proceedings 16, pp. 198–209. Springer, 2017.
>
> [3] Yichang W., et al. Learning interpretable shapelets for time series classification through adversarial regularization. arXiv preprint arXiv:1906.00917, 2019.
>
> > An extensive review ... the best-performing algorithms of the state-of-the-art.
>
> Since our proposed ShapeConv is a deep-learning based new method and some of our baseline have outperformed the state-of-the-art algorithms in [8,9] (e.g. OS-CNN[4] outranking HIVE-COTE[5], WEASEL+MUSE[6] and InceptionTime[7] on both univaraite and multivariate datasets), we only conduct most relevant comparison with shapelet-based and deep-learning based approaches to avoid redundancy. Besides, we add comparison with ROCKET[10] in our revised version.
>
> [4] Wensi T., et al. Omni-scale cnns: a simple and effective kernel size configuration for time series classification. In International Conference on Learning Representations, 2021b.
>
> [5] Jason L., et al. Hive-cote: The hierarchical vote collective of transformation-based ensembles for time series classification. In 2016 IEEE 16th international conference on data mining (ICDM), pp. 1041–1046. IEEE, 2016.
>
> [6] Patrick S., Ulf L.. Multivariate time series classification with weasel+ muse. arXiv preprint arXiv:1711.11343, 2017.
>
> [7] Hassan I. F., et. al. InceptionTime: Finding AlexNet for Time Series Classification. arXiv preprints, arXiv:1909.04939, 2019.
>
> [8] Bagnall, A., et. al. (2017). The great time series classification bake off: a review and experimental evaluation of recent algorithmic advances. Data Mining and Knowledge Discovery, 31(3).
>
> [9] Middlehurst, M., et al. (2023). Bake off redux: a review and experimental evaluation of recent time series classification algorithms. arXiv: 2304.13029
>
> [10] Angus Dempster, et. al. Rocket: Exceptionally fast and accurate time series classification using random convolutional kernels. Data Mining and Knowledge Discovery, pp. 1–42, 2020.
>
> > (Minor comment.) It is considered bad practice to start sentences with mathematical symbols.
>
> Thanks for pointing out. We have revised the paper accordingly.

---

> > ### Author Response · Authors · 2023-11-21
> > **Response to Reviewer VPQr (Part 2)**
> >
> > ### Questions
> >
> > > Convolutional layers are meant to be stacked. Unless I am mistaken, in the experiments, there is only one ShapeConv layer. Would the interpretability of ShapeConv remain if there are several layers?
> >
> > Thanks for your insightful comments! It's true that our ShapeConv can be stacked just like normal convolutional layers to support deep learning. We discussed how we treat ShapeConv as interpretable feature extractor and can be further combined with more sophiscated deep model in Appendix D, and demonstrate its superior performance compared to not using it on large-scale EEG dataset.
> >
> > As for the interpretability, we have added more investigations. Breifly speaking, we have excitedly discovered that a few shapelets are highly similar to waveforms in clinical textbooks. By aligning shapelets with data and conduct shapelet transform, we see clear separatable clusters. The overall experiments in Appendix D demonstrate the possibility of ShapeConv, when stacking with deep models, can benefit medical practitioners in practice by not only offering an accurate judgement, but also pointing out the area of interests with respect to their expert knowledge. **Please kindly refer to the revised version of our manuscript, Appendix D, for figures and more details.**

---

> > > ### Comment · Reviewer_VPQr · 2023-11-23
> > >
> > > I thank the authors for their detailed answers. I still find that the methodological contribution is limited but the amount of experimental validation is very convincing. I have upgraded my rating accordingly.

---

### Author Response · Authors · 2023-11-21
**General Response**

We thank the reviewers for your valuable comments, and we're delighted to hear that our paper is well-written (Reviewer VPQr, JCHN, JpzU, Lmj7) and the experiment is comprehensive (Reviewer VPQr, JCHN, nftB, Lmj7).

One major concern shared by most reviewers is our selection of related works and baselines. Because the time-series classification / clustering task is one of the most essential task in time-series research, there are a considerable number of works coming out every recent year that could be related. Due to the space limitation, we organize our related works as two most relevant parts: Learning Shapelets and Interpretable Time Series Modelling. For the baselines compared in the main experiment section (Sec.4), we choose three kinds of baselines in supervised setting ((1) shapelet-based methods, (2) common deep learning methods, (3) state-of-the-art time-series classification models), and three kinds of baselines in unsupervised setting ((1) pure clustering methods, (2) shapelet-based methods, (3) state-of-the-art time-series clustering models) respectively.

As for more baselines, we leave them in Appendix F. Still, we appreciate the reviews for providing more related materials and we're happy to add them to our paper to make it more comprehensive from more aspects.

Moreover, we appreciate your insightful suggestions which are absolutely helpful for us to polish the paper. Please kindly refer to the updated manuscript with all the major revisions highlighted. The modifications include:
1. Methodology: "Euclidean distance" -> "squared Euclidean distance".
2. Related works: We take the suggestions from the reviewers to add more discussions on some related papers.
3. Experiments:

     a.   Experimental details such as how we choose the subsets to compare, how we obtain results from baselines, how efficient is the initialization step are added.

     b.   Results from more baselines such as ROCKET are included.

     c.   We added more visualization examples for the shapelets learned by ShapeConv.

4. Appendix D, ShapeConv as feature extractor with deep models: we add investigation for interpretability of the learned Shapelets on large-scale dataset.
5. Appendix E, More Visualizations: In E.1, we illustrate the learnt shapelets in 4 additional datasets from UCR. In E.2, we add visualizations for the SHAP Value of the ShapeConv and it ablation.
5. Minor changes on misspellings, inappropriate abbreviations, leading mathematical symbols, etc.

Last but not the least, we would like to express our cheerfulness and excitement that several reviewers recognize our motivation of bridging CNN kernels with shapelets *clear, interesting,* and *fresh*. Researchers have struggled for years to bring the classic shapelet method to the deep learning era in order to achieve a more effective and interpretable learning schema, for example, by coming up with learning-based shapelet discovering algorithms[1][2][3], showing faint but insightful relationship between convolution and shapelet transformation[4][5][6]. And now, we believe our paper together with the ShapeConv is to fill the last piece of the puzzle by **rigorously proving the equivalence** while **offering simple yet effective** solution for the interpretable time-series modeling. Therefore, we hope it can be recognized by the wider ICLR community.

[1] Josif G., et al. Learning time-series shapelets. In Proceedings of the 20th ACM SIGKDD international conference on Knowledge discovery and data mining, pp. 392–401, 2014.

[2] Mit S. et al. Learning dtw-shapelets for time-series classification. In Proceedings of the 3rd IKDD Conference on Data Science, 2016, pp. 1–8, 2016.

[3] Qianli M., et al. Adversarial dynamic shapelet networks. In Proceedings of the AAAI conference on artificial intelligence, volume 34, pp.5069–5076, 2020b.

[4] Arnaud L., et al. Learning dtw-preserving shapelets. In Advances in Intelligent Data Analysis XVI: 16th International Symposium, IDA 2017, London, UK, October 26–28, 2017, Proceedings 16, pp. 198–209. Springer, 2017.

[5] Yeh, C. C. M., et al. (2016). Matrix Profile I: All Pairs Similarity Joins for Time Series: A Unifying View that Includes Motifs, Discords and Shapelets. Proceedings of the IEEE International Conference on Data Mining (ICDM), 1317–1322.

[6] Yichang W., et al. Learning interpretable shapelets for time series classification through adversarial regularization. arXiv preprint arXiv:1906.00917, 2019.

---

> ### Author Response · Authors · 2023-11-22
> **Updates**
>
> Revision:
>     We add the visualization of actual hand movement in Appendix A.2 to verify our interpretations of the GunPoint dataset by shapelets.

---

> > ### Comment · Reviewer_nftB · 2023-11-22
> >
> > I appreciate your efforts in obtaining the original data for gun. And your general responsiveness.
> > --
> > "morphology of waveform in EEG describes its overall shape"
> > This is tautological, the definition of waveform morphology is overall shape.
> > If you meant to define the term, then "The definition of waveform morphology is its overall shape""
> > --
> > "To this end, we visualized a few obtained shapelets out of 2,688 shapelets (128 shapelets per variate, 21 variates in total) and
> > excitedly found that some of them accord with some textbook waveform"
> > Sorry to be skeptical, but if you searched thru  2,688 random examples of EEG, you would have found that some of them accord with some textbook waveform.
> > --
> > I noted
> > "In table E.5 and elsewhere, you report the average accuracy. This is meaningless for datasets of different sizes, class skews, number of classes, default rates etc. To be clear, it is not a flawed metric, it is just meaningless."
> >
> > And you say "We agree that the average accuracy is meaningless for different datasets. However, the community has been using this metric for a long time. We have to report this metric to compare with the previous methods."
> >
> > You agree that some results are meaningless, but you include them?  Why not just add some text to remind the reader that  average accuracy is meaningless?  If you continue the precedent, then the next author will have to report meaningless numbers too. Surely at some point this nonsense must stop.
> > It is almost insulting to assume that "Yeah, WE know this is meaningless stuff, but the reviewers are so dumb they expect it, so we will put it in"
> >
> > --
> > Ultimately, I remain on the fence for this paper.
> > It is nicely written, and the idea of combining the CNN kernels with shapelets is nice.
> > But I was never really convinced by any claims of interpretability.

---

> > > ### Author Response · Authors · 2023-11-23
> > > **Response to Official Comment by Reviewer nftB (part 1)**
> > >
> > > We greatly thank the reviewer for the follow-up comments.
> > >
> > > > "morphology of waveform in EEG describes its overall shape" This is tautological, the definition of waveform morphology is overall shape. If you meant to define the term, then "The definition of waveform morphology is its overall shape"
> > >
> > > "Morphology of waveform in EEG describes its overall shape". You're right, this is exactly a definition. "Morphology of waveform" is an important term in EEG studies, and "overall shape" is a colloquial explanation - though the explanation doesn't exceed ones' expectation when they comprehend directly from the word meaning, which makes it sound tautological. You can check the full sentence [here](https://www.learningeeg.com/terminology-and-waveforms) originating from [1], and similar sentences often comes in the documents the first time "morphology" is mentioned. Thus we prefer not to omit it when referring the whole sentence to explain its importance.
> > >
> > > [1] Lara V Marcuse, Madeline C Fields, and Jiyeoun Jenna Yoo. Rowan’s Primer of EEG. Elsevier
> > > Health Sciences, 2015.
> > >
> > > > “To this end, we visualized a few obtained shapelets out of 2,688 shapelets (128 shapelets per variate, 21 variates in total) and excitedly found that some of them accord with some textbook waveform” Sorry to be skeptical, but if you searched thru 2,688 random examples of EEG, you would have found that some of them accord with some textbook waveform.
> > >
> > > As a matter of fact, we checked only the first ten shapelets learned and we chose the two we recognized immediately as we saw them. This checking process is hard to claim and justify in the paper (and we admit it may due to some luck) so we only provide it here for your information.
> > >
> > > > You agree that some results are meaningless, but you include them? Why not just add some text to remind the reader that average accuracy is meaningless? If you continue the precedent, then the next author will have to report meaningless numbers too. Surely at some point this nonsense must stop.
> > > It is almost insulting to assume that "Yeah, WE know this is meaningless stuff, but the reviewers are so dumb they expect it, so we will put it in"
> > >
> > > We sincerely apologize if anyone feels insulted by our claim, and we have no intention to convey any judgement on anyone reading the statement. We agree that the average accuracy is not that much useful, but we're merely afraid of breaking the convention. But again thanks to the courage you give us, **we have added a footnote in the paper conveying this concern.**

---

> ### Author Response · Authors · 2023-11-23
> **Response to Official Comment by Reviewer nftB (part 2) and Our Thoughts on Time-series Interpretability**
>
> **We are learning from all the feedbacks from the reviewers. Reading your insightful praises and critism, we feel that this is a great opportunity to exchange ideas deeply. Therefore, we're making the following statements to ignite more discussion with the reviewers since we value more on the advances of the whole interpretable machine learning area, a more comprehensive understanding of what we are doing, than the acceptance of this specific paper, while we hope these discussions will not interfere the decision of this paper much.**
>
> > Ultimately, I remain on the fence for this paper. It is nicely written, and the idea of combining the CNN kernels with shapelets is nice. But I was never really convinced by any claims of interpretability.
>
> Again, we thank the reviewer's recognition of our paper. At the meantime, we're curious about the answer to the question, *"What is interpretability? How can a method be treated as a really interpretable and helpful solution that does push the area moving forward"*, from all the readers seeing it. Or more specifically here, *"how we can convince people that our method is really interpretable"*.
>
> The answer may vary from person to person and these are some existing discussions [1,2], and here is the answer we'd like to convey through this paper. We believe that, a truly useful interpretation of a black-box model should be **sensible**, **comprehensible**, and **verifiable** by human. As for time-series classification, such element is the shape of the sequence. As long as a person sees a cliff, a spike, or irregular changes from the plot, he would immediately know "something different is happening". That's why we choose the shapelet as our start point.
>
> Now let me reiterate the motivation of ShapeConv. We found that it's very hard to find the balance between comprehensibility (interpretability) and verifiability (effectiveness). Early shapelet works are easy to comprehend because they come directly from the original data, but can hardly beat contemporary deep models. On the other hand, the learning-based methods can achieve astonishing results, but the shapelets look like random sequences and can hardly provide true insights for human.
>
> We abscribe the gap to the missing of a rigorous eqivalence between shapelets and deep learning. While we found the subtle relationship between convolution and shapelet are about to uncover the mystery, we feel that there is something missing as the finishing touches to make it truly useful. Then comes our ShapeConv - the proof to build the equivalence and several mechanism for applying it to reality.
>
> Here is how we design the experiments to describe why it can be called an interpretable method based on our understanding of interpretability. 1) We visualized many shapelets to make it **sensible**; 2) We explains our shapelet-based findings in real-world datasets and check the human understanding for the question to confirm that it is **comprehensible**; 3) We run tests on a bunch of datasets to prove that it is **verifiable**.
>
> The above statements are only based on our understanding of interpretable time-series classification, which may be controversial from concepts to details. Therefore, we did not include such statements in any part of our paper directly. We list them here because we're eager to learn from your perspective, and we hold the belief that our paper and such discussions will definitely advance the field.
>
> [1] Doshi-Velez F, Kim B. Towards a rigorous science of interpretable machine learning[J]. arXiv preprint arXiv:1702.08608, 2017.
>
> [2] Murdoch W J, Singh C, Kumbier K, et al. Definitions, methods, and applications in interpretable machine learning[J]. Proceedings of the National Academy of Sciences, 2019, 116(44): 22071-22080.

---

### Meta-Review · Area_Chair_tKXp · 2023-12-05

**Metareview:**

The paper proposes a shapelet network with shape and diversity regularizers. The authors highlight a resemblance between a shapelet classifier and a specific form of CNN architecture. The method achieves a high classification accuracy on a wide range of datasets against multiple state-of-the-art baselines. Unfortunately, the authors have not positioned the novelty of the paper well. Instead of proposing a shapelet network with shape and diversity regularizers, and focusing on the fact that such shapelet networks achieve state-of-the-art, the authors have focused on the equivalence between standard convolutional architectures and shapelets. Although the equivalence is interesting, I believe it is a secondary objective compared to demonstrating that the proposed shapelet networks are more competitive than existing shapelet classifiers.

The reviewers were divided and skeptical about the misleading positioning of the paper in demonstrating the equivalence between CNN layers and shapelets. The rebuttal of the authors did not help clarify this point, but in my assessment made the point even more confusing with statements like "We want to emphasize that implementing the equivalence in practice is complex and demands careful design choices, such as specific regularizations for similarity and diversity, and tailored initialization approaches for both supervised and unsupervised learning."

That being said, despite the paper's pitch, the core idea of shapelet networks works well in practice and deserves merits. As a result, I recommend acceptance, conditional to the authors revising the pitch for the camera ready, by downplaying the equivalence and highlighting the results, and the regularization terms. Therefore, I recommend acceptance assuming in good faith that the authors will consider the reviewers' comments for the camera ready.

**Justification For Why Not Higher Score:**

The pitch needs to be reworked to highlight more the fact that the empirical gain comes from the network of shapelets and the two novel regularization terms.

**Justification For Why Not Lower Score:**

The empirical evidence demonstrates the method achieves state-of-the-art predictive performance.

---

### Decision · Program_Chairs · 2024-01-16

Accept (poster)